# Scaffolding Dexterous Manipulation with Vision-Language Models

**Vincent de Bakker**[1,2]     **Joey Hejna**[1]     **Tyler Ga Wei Lum**[1]
**Onur Celik**[2]     **Aleksandar Taranovic**[2]     **Denis Blessing**[2]
**Gerhard Neumann**[2]     **Jeannette Bohg**[1]     **Dorsa Sadigh**[1]
[1]Stanford University     [2]Karlsruhe Institute of Technology

## Abstract

Dexterous robotic hands are essential for performing complex manipulation tasks, yet remain difficult to train due to the challenges of demonstration collection and high-dimensional control. While reinforcement learning (RL) can alleviate the data bottleneck by generating experience in simulation, it typically relies on carefully designed, task-specific reward functions, which hinder scalability and generalization. Thus, contemporary works in dexterous manipulation have often bootstrapped from reference trajectories. These trajectories specify target hand poses that guide the exploration of RL policies and object poses that enable dense, task-agnostic rewards. However, sourcing suitable trajectories—particularly for dexterous hands—remains a significant challenge. Yet, the precise details in explicit reference trajectories are often unnecessary, as RL ultimately refines the motion. Our key insight is that modern vision-language models (VLMs) already encode the commonsense spatial and semantic knowledge needed to specify tasks and guide exploration effectively. Given a task description (e.g., "open the cabinet") and a visual scene, our method uses an off-the-shelf VLM to first identify task-relevant keypoints (e.g., handles, buttons) and then synthesize 3D trajectories for hand motion and object motion. Subsequently, we train a low-level residual RL policy in simulation to track these coarse trajectories or "scaffolds" with high fidelity. Across a number of simulated tasks involving articulated objects and semantic understanding, we demonstrate that our method is able to learn robust dexterous manipulation policies. Moreover, we showcase that our method transfers to real-world robotic hands without any human demonstrations or handcrafted rewards. [1]

## 1   Introduction

Dexterous manipulation is essential for a range of real-world tasks – such as using a power-drill or twisting a door knob – which require the fine-grained control offered by human-like hands [2]. Enabling dexterous capabilities has therefore been a long-standing goal in robotics. Despite the intrinsic advantages of dexterous hands over simpler end-effectors, existing learning paradigms have struggled to cope with their complexity [56]. The prevailing approach for training generalist policies – imitation learning from demonstrations [5, 49] – has achieved limited success with robot hands, primarily due to the challenges of collecting accurate data with dexterous hardware, resulting in a scarcity of high-quality demonstrations [54, 66]. While alternative approaches attempt to re-target demonstrations from easier interfaces [19, 25, 32, 53, 57, 71, 73, 78], e.g., human hands, such approaches often induce irrecoverable errors for fine-grained tasks.

To avoid both data scarcity and the embodiment gap, a combination of reinforcement learning (RL) and sim-to-real transfer has emerged as a promising approach by enabling large-scale experience generation [3]. However, using RL simply shifts the burden from data collection to reward design.

---

[1]Videos at https://sites.google.com/view/vlm-scaffolding/home

39th Conference on Neural Information Processing Systems (NeurIPS 2025).

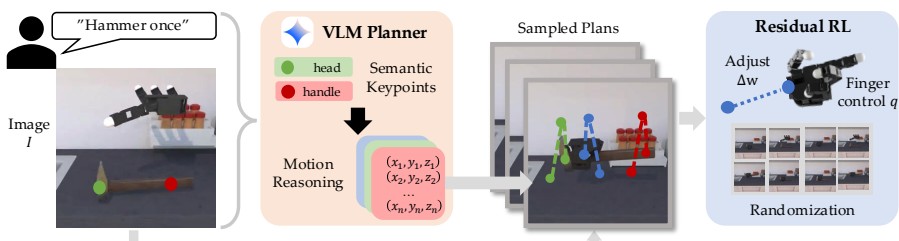

**Figure 1:** Overview of our method: a VLM generates hand and object keypoint trajectories from a language command and scene image. A low-level residual RL policy is trained to track these trajectories in simulation.

Standard RL approaches for dexterous manipulation necessitate hand-crafting complex, task-specific reward functions. A large amount of this complexity arises from the need to guide exploration; with large action spaces, dexterous hands need to be coaxed towards the correct part of the observation space to make progress on a task. Thus, various approaches have used demonstrations to bootstrap the RL process [15, 20, 21, 46, 55, 56]. In dexterous manipulation, this is often done through trajectory tracking, where instead of designing a complex reward function, a policy is rewarded for tracking the exact wrist and object positions in a demonstration, leaving RL to only make adjustments [6, 79]. By re-framing dexterous manipulation as a trajectory tracking problem, such approaches can leverage dense, task-agnostic rewards and guide exploration by using residual policies [16, 26].

Though demonstration tracking overcomes the design challenges associated with RL, it paradoxically re-introduces the same dependence on demonstrations we sought to avoid in the first place. For example, prior works [6, 79] that use tracking-based RL for dexterous manipulation often require large prior datasets with thousands of teleoperated demonstrations [13, 76], restricting the method to tasks for which data has already been collected. Furthermore, the dependence on accurate reference trajectories extends to test time, meaning that either new demonstrations are needed each time the environment changes, or a high-level policy must accurately predict them. In the context of prior work, either solution requires more demonstrations. However, for the majority of practical tasks it is unclear why demonstrations were chosen as reference trajectories when RL only requires that motions are accurate enough to provide a suitable reward and guide exploration.

Our key insight is that coarse motion plans ("scaffolds") can be sufficient for both of these criteria. Generating such plans only requires high-level spatial and semantic reasoning, the exact abilities afforded by new advancements in vision-language models (VLMs). Due to their training on vast and diverse datasets encompassing real-world knowledge and embodied reasoning [64], VLMs have been able to synthesize [14, 44] and discriminate [24, 48] desired robot motions. Consequently, VLMs have the potential to supply the high-level reward signals and exploratory guidance needed for RL through coarse motions. So long as these motions generally encapsulate the desired behavior, RL can optimize per-timestep offsets and finger motions to maximize the tracking reward, ultimately surpassing human teleoperation in both performance and precision, eliminating the reliance on demonstrations.

Building upon this insight, we introduce a framework for learning manipulation policies for dexterous robot hands with VLM-generated motion plans and residual RL. Given a natural language instruction (e.g., "hammer once" Fig. 1) and image, an off-the-shelf VLM first identifies relevant object keypoints. Then, provided the initial keypoints and hand pose, the VLM generates the associated 3D trajectories for both object and hand motions to define the supervision targets for a "low-level" residual RL policy trained in simulation. By controlling the robot's hands and fingers, the low-level policy learns to effectively track the trajectory and complete the desired task. Using VLMs for high-level guidance has a number of distinct benefits. Through repeated querying, we can randomize the initial keypoints and high-level trajectories, enabling generalization to new unseen initial conditions and new trajectories at test-time. Moreover, in situations where a VLM's high-level planning is error-prone, performance can often be substantially improved by simply providing additional in-context examples.

We evaluate our method across a suite of challenging dexterous manipulation tasks in simulation requiring semantic understanding, human knowledge about concepts like "hammering", and precise manipulation for difficult or articulated objects. Across 8 tasks, our method achieves close performance in both success rate and generalization to handcrafted, oracle plans despite requiring no manual reward engineering. We demonstrate successful sim-to-real transfer on a physical dexterous robot, achieving robust performance without any human demonstrations or manually designed rewards.

## 2   Related Work

**Planning with Vision-Language Models.**  Recent advances have demonstrated the potential of VLMs to guide robotic planning through their powerful semantic and spatial reasoning capabilities. One family of approaches directly synthesizes policies by translating natural language instructions into executable code using low-level perception and control APIs [23, 34, 59, 72]. To extend this to dexterous manipulation, [36] integrates predefined skill libraries, at the expense of limiting generalization and behavioral diversity.  Other efforts propose using VLMs to plan actions by generating spatial keypoint constraints [22] or directly producing waypoints [14, 50]. However, such methods operate in an open-loop fashion and lack the closed-loop feedback necessary for fine-grained, adaptive control in dexterous tasks. Aside from directly learning policies, several works use VLMs to code dense reward functions [41, 65, 80], but these approaches often require privileged access to environment internals and result in opaque and sometimes hard-to-interpret reward structures. Other approaches more directly leverage the vision capabilities of VLMs to act as success detectors [12, 30, 75], reward functions [42], or value functions [40, 77] for RL. Oftentimes, these quantities can be learned from VLM generated preferences [28, 31, 69]. However, all of these approaches are often too imprecise to produce the dense optimization signals required for dexterous manipulation and are less efficient than using the VLM to simply produce a plan.

**Learned Dexterous Manipulation.** Though early works demonstrated the feasibility of deploying in-hand manipulation policies trained in simulation on real robots [3, 7, 18, 35], they relied on carefully crafted reward functions for each task. Such approaches have proven most successful in locomotion [1, 4, 29], where rewards are more easily designed and terrain can be replicated, unlike object dynamics in manipulation. More recent efforts scale to full-arm dexterity and multi-object grasping [39, 60], while others incorporate human priors to improve sample efficiency [43]. Despite these advances, most approaches are still limited to only a set task, e.g. object grasping or rotation [52, 67], where manually, task specific rewards can be designed. However, this approach remains inherently unscalable to more complex and non-cyclic tasks. Prior works address this bottleneck by tracking motions from demonstrations [6]. Our work aligns with this shift, but instead sources coarse motions from VLM feedback.

**Dexterous Manipulation by Tracking Motions.**  When framing dexterous manipulation as a tracking problem, dense rewards are easy to obtain via tracking error [4, 15, 51].  Some systems leverage motion capture data to extract object and wrist trajectories from human demonstrations, which are then used to train tracking policies in simulation via residual RL [6, 33] or through behavior cloning [17]. Other approaches improve robustness by iteratively adding successful rollouts to the training dataset [37].  Recent work also shows that a single demonstration can bootstrap effective policy learning [15, 38]. However, all of these methods depend on human demonstrations, which are expensive to collect and difficult to scale. Moreover, policies trained on such data often fail to generalize to novel initial states. Our method retains the advantages of the trajectory tracking framework – dense supervision and residual policy learning – while eliminating the dependency on demonstrations by using VLMs to infer target trajectories directly.

## 3   Dexterous Manipulation via VLM Feedback

We focus on dexterous manipulation using robotic hands with visual observations and natural language instructions, with the aim of developing a general approach transferable across diverse applications and settings. Following prior work [4, 6], we adopt a hierarchical approach that naturally delineates planning and control.  However, instead of centering plans around demonstrations, we leverage a VLM to produce coarse plans sufficient to "scaffold" low-level RL. We interface between these two components using 3D keypoints, as they provide sufficient precision for effective manipulation [68, 74], yet are abstract enough for VLM reasoning [14, 44] and often used during pre-training [27, 64].

### 3.1   Problem Formulation

Our goal is to learn a hierarchical policy for dexterous manipulation, where the high- and low-level policies interface via 3D keypoint-based plans or trajectory "scaffolds". While several prior works assume access to ground-truth states (often in simulation), such information is typically only partially observable in practice. For example, it is unrealistic to assume that one is able to precisely measure object poses and velocities in the real world. Only the dexterous hand's proprioceptive state $(\mathbf{w}, \mathbf{q}, \dot{\mathbf{q}})$

comprised of the current wrist pose $\mathbf{w} \in \mathrm{SE}(3)$, finger joint positions $\mathbf{q}$ and velocities $\dot{\mathbf{q}}$ is exactly known. Instead of ground-truth states we assume access to RGB images $I$, depth $D$, and a language instruction $L$ which communicates the task. Following standard practice in dexterous manipulation, we use an absolute action space comprised of desired wrist $\mathbf{w}^{\mathrm{targ}}$ and finger joint positions $\mathbf{q}^{\mathrm{targ}}$, i.e., $(\mathbf{w}^{\mathrm{targ}}, \mathbf{q}^{\mathrm{targ}}) \in \mathcal{A}$.

The high-level policy $\pi^h$ produces a coarse, 3D keypoint-based plan $\tau$ from the language instruction $L$ and an initial high-level observation $o_1^h$ at time $t = 1$ containing the initial image $I_1$ and wrist position $\mathbf{w}_1$. As we instantiate $\pi^h$ using a VLM, we assume the ability to project 2D keypoints $\mathbf{u}^{(i)} \in \mathbb{R}^2$ in image space to 3D keypoints $\mathbf{x}^{(i)} \in \mathbb{R}^3$ in world coordinates, which is easily accomplished in practice using depth information $D$ and camera parameters (intrinsic and extrinsic). The number of 3D keypoints $k$ in the final plan $\tau$ is specified through the instruction $L$. We enumerate these keypoints as $\mathbf{x}^{(1)}, \ldots \mathbf{x}^{(k)}$ and abbreviate sequences of length $T$ through time via the short-hand $1 : T$. The final keypoint plan $\tau$ includes $k$ 3D keypoint sequences $\mathbf{x}_{1:T}^{(1)}, \ldots \mathbf{x}_{1:T}^{(k)}$ and a sequence of predicted wrist poses $\tilde{\mathbf{w}}_{1:T}$. This coarse plan encapsulates both information about the task via the $k$ keypoint sequences which can capture object movements, and information to guide the agent's exploration via the wrist position $\mathbf{w}$. The high-level policy can be written as:

$$\pi^h(\underbrace{\tilde{\mathbf{w}}_{1:T}, \mathbf{x}_{1:T}^{(1)}, \ldots \mathbf{x}_{1:T}^{(k)}}_{\tau} \mid \underbrace{I_1, \mathbf{w}_1}_{o_1^h}, L) \tag{1}$$

The high-level policy only provides a coarse plan for the wrist $\mathbf{w}$ – not the finger joint positions $\mathbf{q}$ which will be learned by the low-level policy with RL.

The low-level policy $\pi^l$ produces wrist and finger actions $a_t$ to execute the keypoint plan $\tau$. We assume access to a keypoint tracking model, which given an initial 3D keypoint $\mathbf{x}_1^{(i)}$ at time $t = 1$ is able to track its position over time to produce estimates $\hat{\mathbf{x}}_t^{(i)}$. The low-level policy $\pi^l$ is then optimized via RL using a reward function that encourages consistency between the estimated 3D keypoints $\hat{\mathbf{x}}_t^{(i)}$ and those produced by the plan $\tau$, $\mathbf{x}_t^{(i)}$. To accomplish this task, it takes as input both a low-level observation $o_t^l \in \mathcal{O}^l$, consisting of the proprioceptive state $(\mathbf{w}, \mathbf{q}, \dot{\mathbf{q}})$ and estimated keypoints $\hat{\mathbf{x}}^{(i)}, \ldots, \hat{\mathbf{x}}^{(i)}$, and all future steps of the plan $\tau_{t:T}$. Succinctly,

$$\pi^l(\underbrace{\mathbf{w}_t^{\mathrm{targ}}, \mathbf{q}_t^{\mathrm{targ}}}_{a_t} \mid \underbrace{\mathbf{q}_t, \dot{\mathbf{q}}_t, \mathbf{w}_t}_{\mathrm{proprio}}, \underbrace{\hat{\mathbf{x}}_t^{(1)}, \ldots, \hat{\mathbf{x}}_t^{(k)}}_{\mathrm{keypoint\ estimates}}, \underbrace{\tilde{\mathbf{w}}_{t:T}, \mathbf{x}_{t:T}^{(1)}, \ldots, \mathbf{x}_{t:T}^{(k)}}_{\mathrm{plan}\ \tau_{t:T}}). \tag{2}$$

Provided the high- and low-level decomposition of our approach, we now describe each component.

## 3.2 Trajectory Generation for High-Level Policies via VLMs

We implement the high-level policy $\pi^h$ using a VLM, which must be able to effectively translate the task description $L$ and initial image $I_1$ into a coarse motion plan $\tau$ for $\pi^l$ to complete. This necessitates a high-degree of semantic and spatial reasoning capabilities: the paths of different keypoints $\hat{x}_{1:T}^{(i)}$ must obey the desired relationships between objects (e.g. put the apple on the cutting board) and physical constraints (e.g. the head of the hammer must remain attached to the handle). At the same time, the predicted wrist trajectory $\tilde{\mathbf{w}}_{1:T}$ must remain close to the objects of interest to facilitate manipulation. We generate coarse keypoint plans $\tau$ using VLMs in three phases: (1) semantic keypoint detection, (2) coarse trajectory generation, and (3) interpolation. The left of Fig. 2 provides a visual overview. Optionally, generated plans $\tau$ can be improved using few-shot prompting.

**Keypoint Detection.** First, we elicit task-relevant keypoint names from the VLM using a standardized preamble. For example, for the "hammering" task, these keypoints (Fig. 1) could include both the handle and head of a hammer, or the position of an object and its desired location for semantic pick-place (Fig. 2). The VLM then identifies $k$ 2D keypoints $\mathbf{u}^{(1)}, \ldots, \mathbf{u}^{(k)}$ in the image $I$ that are relevant to completing the task described via text $L$. The VLM is prompted with useful keypoints for the task. The full prompts used are included in Appendix H. Since the VLM operates in the 2D image plane, we lift 2D keypoints $\mathbf{u}$ to 3D world coordinates $\mathbf{x}$ using depth information.

**Trajectory Generation.** Second, provided the text description $l$, the VLM generates waypoint sequences of length $n < T$ for each of the initial 3D keypoints $\mathbf{x}^{(1)}, \ldots, \mathbf{x}^{(k)}$ and the wrist position

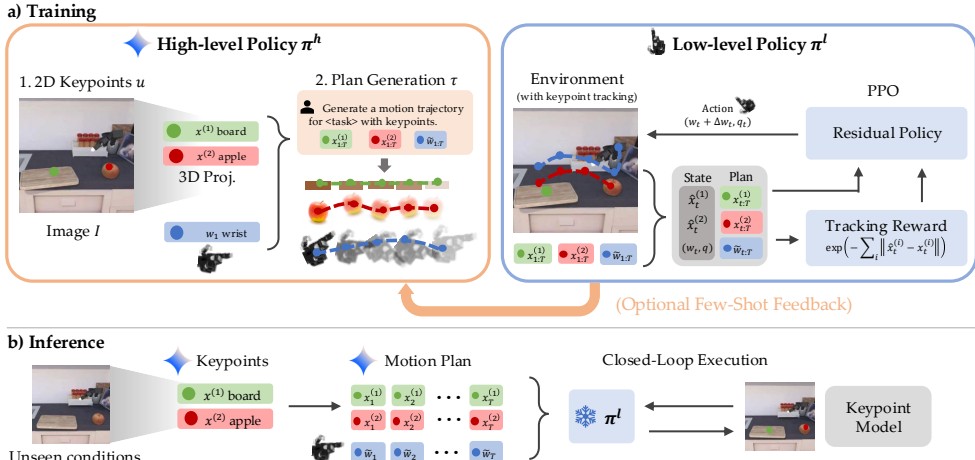

**Figure 2:** a) Training: a high-level VLM predicts 3D keypoint plans, which a low-level policy learns to track via RL. b) Inference: new plans are generated by the VLM, which are executed by the frozen low-level policy.

$\mathbf{w}_1$. In total, this results in $(k + 1) \times n$ 3D waypoints which will serve as the basis of the plan $\tau$. We include the full prompts used in Appendix H. While the first keypoint detection stage depends on the VLM's image understanding, this phase depends more on spatial understanding and reasoning – the VLM must translate semantic descriptions into motions, e.g., what "hammering" implies or how a door opens, while respecting the physical constraints between keypoints and proximity between the hand and manipulated objects. Note that we do not have the VLM produce keypoint trajectories of the full horizon $T$, as doing so might be more difficult and inaccurate. Instead, we posit the quality of each waypoint matters more than the number, as low-level RL can compensate for small mistakes in position but not large errors in reasoning.

**Interpolation.** Finally, though we have coarse waypoint trajectories of length $n$ for all keypoints and the agent's wrist pose, directly using these waypoints as motion targets may result in excessively fast or jittery motion. Thus, we additionally apply linear interpolation to convert the $n$ waypoints into sequences of length $T$, e.g. $\mathbf{x}_{1:T}^{(i)}$, to form the final plan $\tau$ used for training the low-level policy $\pi^l$.

**Few-Shot Improvement.** Though VLM-generated keypoint plans $\tau$ are often correct, they are not infallible. For example, sometimes the high-level policy $\pi^h$ will flip the world-coordinate axes, resulting in unintelligible keypoint plans. Such errors from the VLM plan are irrecoverable if they fail to complete the task. However, the accuracy of VLMs can often be improved by providing in-context examples [11, 40]. After deploying the final system as described later in Section 3.4, we can use examples of plans successfully executed by the low-level policy as in-context examples for future generations. Provided $m$ successful plans $\tau^{(1)}, \ldots, \tau^{(m)}$, we can prompt the high-level policy as $\pi^h(\tau|s_1, \tau^{(1)}, \ldots, \tau^{(m)})$ to produce better plans for the low-level policy. As shown in our experiments, iteratively repeating this procedure can further increase performance as the in-context plans improve.

### 3.3 Low-Level Control with Reinforcement Learning

The low-level policy $\pi^l$ ensures that the keypoint plan $\tau$ provided by $\pi^h$ is effectively tracked. We learn $\pi^l$ using residual reinforcement learning [16, 26], which we formalize through a "plan" conditioned MDP on top of the low-level observation space $\mathcal{O}^l$ and action space $\mathcal{A}$ with horizon $T$. We assume the dynamics to be stochastic $p(o_{t+1}|o_t, a_t)$ to account for noise in keypoint estimation and that the initial state $o_1^l \sim p_\tau^{\text{init}}$ is always consistent with the high-level plan $\tau$ to ensure its validity. Naïvely, $\pi^l$ is optimized to maximize the expected cumulative reward provided plans sampled from $\pi^h$, $\max_{\pi^l} \mathbb{E}_{\tau \sim \pi^h(\cdot|o_1^h)} \mathbb{E}_{o_{1:T}^l \sim \pi^l(\cdot|\tau)}[\sum_{t=1}^T r_\tau(o_t^l)]$ where $\pi^l(\cdot|\tau)$ represents the distribution of full trajectories of length $T$ under $\pi^l$ and $p_\tau^{\text{init}}$. In this section, we describe how we use the plan $\tau$ to further guide the learning and exploration of $\pi^l$ through the reward function, policy parameterization, and environment termination conditions (right half of Fig. 2).

**Dense Keypoint Rewards.** Standard RL based approaches for dexterous manipulation often require complex, hand-crafted reward functions. However, provided a high-level keypoint plan $\tau$ dictating

how all objects should move and interact, we can simply reward the agent for following the plan via keypoint distances. Though similar ideas have been used for tracking reference demonstrations [6] with ground-truth object poses, we instead track keypoints, which do not require full observability. Our final reward function is given by

$$r_\tau(o_t) = \underbrace{\exp\left(\frac{-\beta}{k}\sum_{i=1}^{k}\|\hat{\mathbf{x}}_t^{(i)} - \tilde{\mathbf{x}}_t^{(i)}\|_2\right)}_{\text{Keypoint Tracking}} + \underbrace{\exp\left(-1/(N_{\text{contact}}(o_t) + \epsilon)\right)}_{\text{Maintaining Contact}}, \tag{3}$$

where the first term is a function of the mean Euclidean distance between the planned and observed keypoint positions, and the second term $N_{\text{contact}}(o_t)$ represents the number of finger tips in contact with the environment. Rewarding contact (second term) incentivizes stable grasping, while the first term encourages accurate tracking of the plan. This reward formulation is significantly simpler than those used in previous RL approaches that lack trajectory supervision [39, 60] and can be applied to any task sufficiently captured by keypoint trajectories. We do not reward the policy for tracking the wrist $\tilde{\mathbf{w}}_{1:T}$ to allow it to adjust as needed. Instead, we use $\tilde{\mathbf{w}}_{1:T}$ in the policy parameterization itself.

**Residual Policy.** To guide the agent towards the objective specified by the high-level plan $\tau$, we employ "residual" RL [16, 26] in the absolute pose action space $\mathcal{A}$. Specifically, the learned low-level policy $\pi_\theta^l$ predicts offsets $\Delta\mathbf{w}$ to the wrist plan $\tilde{\mathbf{w}}_t$ instead of absolute actions $\mathbf{w}^{\text{targ}}$. Mathematically, this can be written as follows:

$$a_t = (\tilde{\mathbf{w}}_t + \Delta\mathbf{w}, \mathbf{q}_t^{\text{targ}}), \text{ where } (\Delta\mathbf{w}, \mathbf{q}_t^{\text{targ}}) \sim \pi^l(\cdot|o_t). \tag{4}$$

This guarantees that the learned policy follows the plan's wrist trajectory $\tilde{\mathbf{w}}_{1:T}$ by default and clipping of $\Delta\mathbf{w}$ ensures it never deviates too far. This residual approach uses the world knowledge encoded by the VLM plan to guide exploration of the low-level policy to relevant parts of the state space for completing the objective. Practically, $\pi^l$ is implemented as a multi-layer perceptron where keypoints are provided in a fixed order and future planning steps $\tau_{t:T}$ are down-sampled to a fixed length.

**Termination Conditions.** To improve learning efficiency, we terminate episodes early if the tracking error, $\frac{1}{k}\sum_{i=1}^{k}\|\hat{\mathbf{x}}_t^{(i)} - \tilde{\mathbf{x}}_t^{(i)}\|_2$, exceeds a threshold $\delta$. This early stopping criterion serves as a strong supervisory signal, encouraging the policy to remain close to the intended trajectory. To further guide learning, we introduce a curriculum: the initial threshold $\delta_{\text{init}}$ is linearly annealed to $\delta_{\text{init}}/2$ over the course of training. This facilitates broad exploration in the early stages while promoting precise trajectory tracking later on. We select task-specific values for $\delta_{\text{init}}$, provided in Appendix B.3.

## 3.4 The Full Pipeline

The aforementioned components define the process of generating a single plan $\tau \sim \pi^h(\cdot|s_1)$ and using it to learn a low-level residual policy $\pi^l$. Here we describe our full training and inference pipeline.

**Training.** The final low-level policy must be able to perform well across all plans generated by $\pi^h$, which can differ in initial conditions, the selected keypoint locations, and the generated motions. Thus, following the objective from Section 3.3, we train the full system on variations in the initial poses of objects the dexterous hand. For each of $N$ initial conditions from the environment, we sample corresponding high-level plans from $\pi^h$. We then train the low-level policy using PPO [58] by randomly sampling from the set of $N$ initial conditions and plans across massively parallelized simulation environments. In simulation, we track keypoints using ground-truth object information to generate low-level observations $o^l$. Training across randomized plans is crucial for $\pi^l$ to be robust to both the keypoints and plans generated by $\pi^h$. Further training details and hyperparameters are in Appendix A.

**Evaluation.** At test time (Fig. 2 b)), we randomize the initial conditions of the environment. Afterwards, we generate a new plan using the VLM $\pi^h$ and supply it to the frozen learned policy $\pi_\theta^l$ for closed loop control. The high-level policy inherits the robustness of the underlying VLM to visual perturbations, allowing it to easily transfer to the real world. We generate $\tau$ using captured RGB-D images, and deploy the low-level policy zero-shot in the real world. We estimate the keypoint positions for low-level observations $o^l$ using pose estimators [70].

## 4 Experiments

We conduct a comprehensive suite of experiments to assess the effectiveness, generality, and robustness of our method across a diverse range of dexterous manipulation tasks. Our evaluation is

structured around four core questions: 1) Are VLM scaffolds effective for a broad range of dexterous tasks? 2) How much can iterative refinement improve performance? 3) What causes VLM scaffolds to fail? 4) Can our method successfully learn policies that transfer to the real world?

## 4.1 Experimental Setup

**Task Suite** We construct an evaluation suite using the ManiSkill simulator [45, 62] and Allegro Hand model designed to evaluate four core dexterous manipulation capabilities for which motion planning is difficult: i) semantic understanding, ii) unstructured motion, iii) articulated object manipulation, and iv) precise manipulation. Each of the eight tasks, two per category, is depicted in Fig. 3. Instead of reward functions, each task is specified by a language instruction $L$. For example, the instruction for the "Move Apple" task is "Move the apple to the cutting board". The high-level VLM $\pi^h$ is additionally guided by a prompt to detect specified keypoints. Further details can be found in Appendix F. Crucially, the capabilities evaluated by our task set are difficult to design reward functions for (articulated object manipulation or requiring complex and unstructured motion) or are challenging to specify using classical motion planning (requiring semantic knowledge or precision).

**Methods** Given the novelty of our problem setting, there are few applicable baselines which are also language-conditioned, demonstration-free, and do not require ground-truth state estimation. Thus, we mainly focus our experiments on comparison with a variety of oracles and ablations:

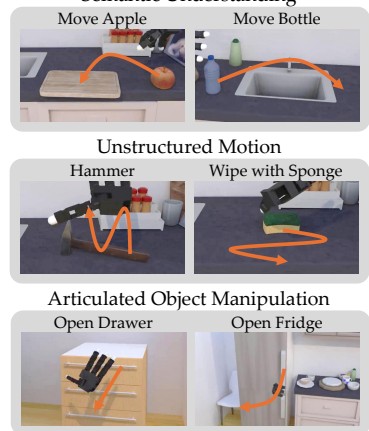

Semantic Understanding
Move Apple        Move Bottle

Unstructured Motion
Hammer        Wipe with Sponge

Articulated Object Manipulation
Open Drawer        Open Fridge

Precise Manipulation
Close Scissors        Close Pliers

- **Iterative Keypoint Rewards (IKER):** We implement Iterative Keypoint Rewards [50], in which a VLM generates code specifying reward-function parameters. We use the same prompting procedure and setup as in our method for a fair comparison. Since IKER assumes a fixed set of keypoints, we adopt the same VLM-identified keypoints used by our system to ensure parity.
- **Pre-recorded Trajectories:** This method reuses pre-recorded trajectories from the training set at test-time, eliminating adaptability to new scenarios.
- **Oracle Keypoints and Trajectories:** This baseline uses fixed, manually defined keypoints and hard-coded trajectories for each task, representing an upper bound on performance with perfect semantic understanding and keypoint detection.
- **Reduced Waypoints:** We artificially constrain the VLM to produce shorter waypoint sequences, e.g., length three instead of $n = 20$, reducing the complexity of motion that can be expressed via the keypoints and wrist.

**Figure 3:** A depiction of the eight tasks used for evaluation. Each task belongs to one of four overarching categories.

We compare against additional reinforcement learning and imitation learning baselines and additionally ablate adding systematic noise into VLM predictions in Appendix E

We evaluate two versions of our system: a zero-shot variant, where the vision-language model (VLM) receives no example plans, and a few-shot variant, where it is provided with three examples of successful plans $\tau$ in-context (Section 3.2).

**Architectures.** We use Gemini 2.5 Flash Thinking [63] as the high-level policy with a thinking budget of 1000 tokens for plan generation. A discussion on the use of open-source VLMs is provided in Appendix D.

The low-level policy $\pi^l$ is implemented as a 3-layer MLP with hidden dimensions of size 512 and ELU activations [9]. We sample 100 initial states and corresponding plans $\tau$ for training $\pi^l$ with PPO [58].

**Evaluation.** For evaluation, we construct task-specific binary success metrics (e.g., object reaches target position, door opens to a minimum angle) to measure performance. All policy evaluations are conducted across 100 initial states with novel object configurations and hand poses. We run 20 trials for each configuration for a total of 2000 evaluation episodes and average results across three seeds.

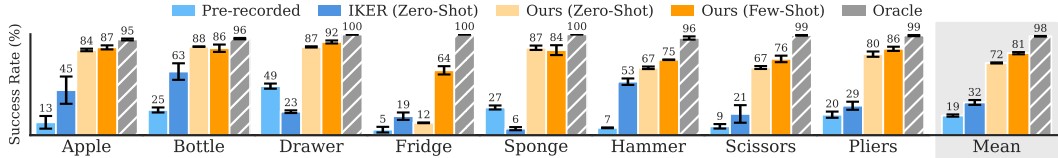

**Figure 4:** Results on the simulation task suite. Success rate (in %) is averaged across 3 seeds and uncertainty is given by the standard error. Our method performs nearly as well as the oracle with perfectly scripted plans.

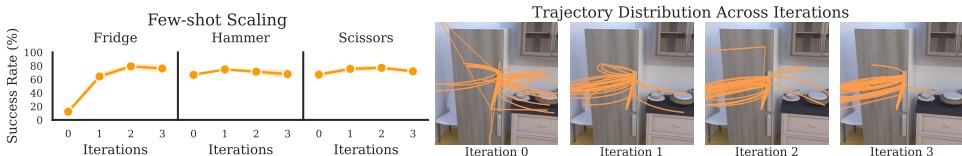

**Figure 5:** (Left) The performance of our method as we iteratively refine the high-level policy $\pi^h$ by providing successful plans $\tau$ in-context. (Right) The projected 3D plans on the evaluation set for each iteration.

**Simulation Results.** Fig. 4 shows the success rates for the different simulation tasks. Our method with few-shot adaptation achieves consistently high success rates, with an average success rate of 72%, often approaching the performance of the oracle with perfect scripted plans. This suggests that modern VLMs are capable planners for scaffolding dexterous policies. Moreover, we observe that adding a few successful examples can further improve performance beyond that of zero-shot in six out of eight tasks. The most notable improvement is from the Fridge task, where the VLM commonly flips the world axes, resulting in implausible plans. Thus, providing examples of successful plans drastically improves performance. Performance of the pre-recorded baseline remains poor for all tasks, except in the drawer task, where the novel plans are likely less important at test time due to the sizable width of the drawer handle.

**Iterative Refinement.** We provide the VLM with successful trajectories from the training set as in-context examples to improve the proposed waypoints. We iterate this process up to three times in Fig. 5. All tasks show an increase in the success rate after the first iteration, with diminishing returns after the second iteration. This is likely because plan-generation performance plateaus and errors in other parts of the system dominate. As before, the most significant improvement is in the Fridge task. While VLMs already have solid semantic understanding, few-shot iteration can improve performance and correct common mistakes during inference. After iterative refinement, the overall success rate improves to 81%.

## 4.2 What Causes VLM Scaffolds To Fail?

**Failure Modes.** To comprehensively evaluate the failure modes of our pipeline across all tasks, we present a Sankey diagram in Fig. 6, categorizing errors into three primary sources: (i) incorrect keypoint detection, where keypoints do not lie on target objects, indicating deficiencies in VLM keypoint detection; (ii) incomplete trajectory tracking by the low-level policy, suggesting either inaccuracies in the low-level policy or unsuitable trajectories; and (iii) tracked trajectories that nonetheless fail to achieve success, revealing shortcomings in VLM trajectory generation.

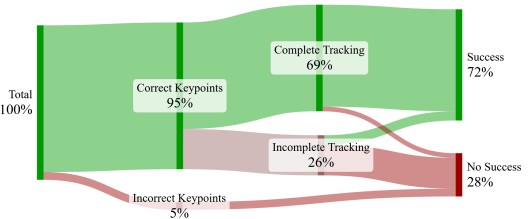

**Figure 6:** Error decomposition across failure cases. Most errors stem from incomplete trajectory tracking, followed by keypoint detection issues.

Notably, (iii) can occur because success criteria are independent of tracking, allowing fully tracked trajectories to fail. Our analysis reveals that the most significant failure mode is incomplete trajectory tracking, occurring in 26% of the rollouts. This suggests a critical bottleneck in the low-level policy's ability to follow the planned path, though it remains unclear whether this stems from unrealistic trajectory proposals or intrinsic policy limitations. In contrast, keypoint detection errors account for 5% of failures, while unsuccessful but fully tracked trajectories contribute 3% (not shown in diagram). Interestingly, some partially tracked trajectories still succeed, implying that perfect tracking is not always required. These findings highlight the importance of improving both trajectory generation and

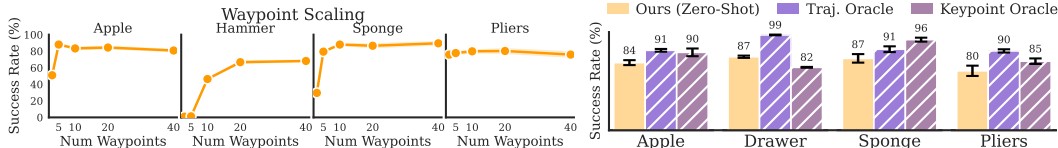

**Figure 7:** (Left) Task success vs. number of waypoints in VLM plans. Most tasks saturate by 10 waypoints; only the hammer task benefits from denser trajectories. (Right) Ablation of VLM components. Replacing keypoints or trajectories with oracles highlights their relative impact across tasks.

tracking robustness. However, further automated decomposition of these errors remains challenging due to the complex interplay between high-level planning and low-level execution.

**Number of Waypoints.** In Fig. 7 (Left), we evaluate the performance of our method using 3, 5, 10, 20, and 40 waypoints for plan generation. The results show that planning fidelity is typically not a large source of error, unless very few (3 or 5) waypoints are used. In 3 of 4 tasks, performance saturates with 10 waypoints. Only the hammer task requires 20, as it needs repeated motion.

**VLM Components.** To ablate the impact of using a VLM for keypoint detection and plan generation, we replace each component with an oracle in Fig. 7 (Right). For the Keypoint oracle, we use hand-specified keypoints for generating $\tau$. For the Traj. oracle, we use VLM keypoints but script plans for $\tau$. The resulting improvements vary across tasks: in the drawer task, the Traj. oracle achieves near perfect performance indicating planning was the bottleneck, however, in the sponge task the keypoint oracle improves performance the most, indicating that $\pi^h$ most commonly mis-specifies keypoints.

## 4.3 Real-World Results

To evaluate sim-to-real transfer, we deploy our system on a real robot using the same inference pipeline as in simulation. From a single real-world RGB-D image and a natural-language command, the vision-language planner generates wrist and keypoint trajectories. The low-level policy is trained entirely in simulation using a digital twin of the real-world environment, and then executed in the real-world, conditioned on the generated trajectories. Robustness to visual distractors is achieved by training the policy on state-based observations, decoupling it from raw visual input. Robustness to discrepancies in physical parameters is achieved through domain randomization in simulation. Our real-world experiments are performed using a 16-DoF Allegro hand mounted on a 7-DoF KUKA LBR iiwa 14 arm in a tabletop setting, and a ZED 1 stereo camera rigidly mounted to the table.

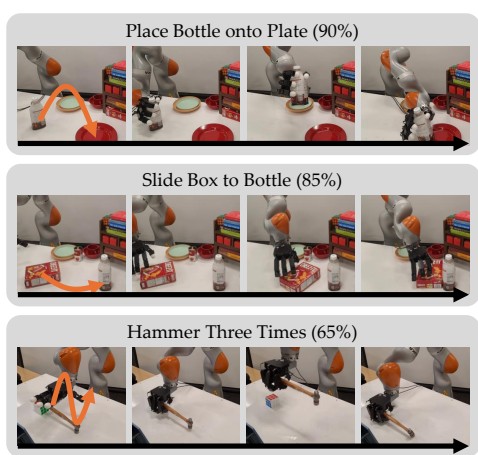

**Figure 8:** Real-world rollouts of **Place Bottle onto Plate**, **Slide Box to Bottle** and **Hammer Three Times**.

We evaluate three real-world manipulation tasks: **Place Bottle onto Plate**, **Slide Box to Bottle**, and **Hammer Three Times**, covering semantic placement, non-prehensile pushing, and complex motion planning (Fig. 8). Each task is executed for 20 rollouts. Our system achieves a 90% success rate on **Place Bottle onto Plate**, 85% on **Slide Box to Bottle**, and 65% on **Hammer Three Times**. These results indicate that the proposed modular, trajectory-based framework generalizes effectively from simulation to the real world, maintaining robust performance across a range of manipulation settings. Additional implementation and hardware details are provided in Appendix C.

## 5 Conclusion

We presented a new framework for dexterous robotic manipulation that combines VLMs with reinforcement learning to generate and execute semantically meaningful hand-object trajectories. By casting manipulation as a trajectory-tracking problem using VLM-generated keypoint plans,

our method eliminates the need for human demonstrations or handcrafted reward functions, while enabling generalization across diverse objects, goals, and scene configurations.

Our experiments in both simulation and the real world show that this approach reliably solves a variety of complex manipulation tasks, including articulated objects, semantic reasoning, and fine finger control. The system exhibits strong generalization to novel keypoints and configurations, and transfers effectively to physical hardware without additional tuning or data collection.

**Limitations and Future Work.** While our method demonstrates strong performance across a range of tasks, several limitations remain along with exciting directions for future work. First, our current approach assumes rigid-body interactions, which simplifies keypoint tracking but limits applicability to deformable objects. Extending to more complex, deformable objects would require the ability to track keypoints on non-rigid surfaces (e.g., point tracking models [27]). Additionally, while our approach can implicit infer orientation from three non-colinear keypoints, this is less effective for smaller objects with fewer keypoints. We see potential in exploring richer scaffold representations and more advanced VLMs to better capture orientation in these cases. The high-level trajectory generation in our system currently does not account for the low-level controller's capabilities, which limits adaptability. Integrating feedback from the low-level controller into the VLM planner to form a more closed-loop system offers an exciting opportunity for improvement. Additionally, the current reasoning time of 1-2 minutes for VLMs restricts real-time responsiveness, motivating a shift towards faster, more efficient models. While our scaffold-based reward design is lightweight and scalable, we see opportunities to combine it with additional task-specific signals, such as force or torque constraints, to enhance performance in more complex tasks. Although the system shows strong zero-shot generalization across various object configurations, generalizing to new tasks or object categories remains a challenge. We believe combining scaffold-based reasoning with object-agnostic representations could unlock broader adaptability. Lastly, sim-to-real transfer remains an ongoing challenge, especially for tasks requiring high-precision control. We see exciting directions for improving simulation realism, multi-view perception, and sensor fusion to narrow the sim-to-real gap, particularly for fine-grained tasks such as manipulation.

### Acknowledgments

We would like to thank Marcel Torne Villasevil, Michelle Pan, Shuang Li, Max Du, Amber Xie, Suvir Mirchandani, and Priya Sundaresan for their important feedback on the paper draft. This work is supported by the National Science Foundation under Grant Numbers 2342246, 2006388, 2132847 and 2218760, and by the Natural Sciences and Engineering Research Council of Canada (NSERC) under Award Number 526541680. This work is additionally funded by the German Research Foundation (DFG, German Research Foundation) - SFB-1574 - 471687386, from the pilot program Core Informatics of the Helmholtz Association (HGF), and supported by the European Research Council (ERC) under the European Union's Horizon Europe programme through the project SMARTI³ (Grant Agreement No. 101171393).

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

# A Hyperparameters

## A.1 PPO

The hyperparameters of our PPO training are detailed in Table 1.

**Table 1:** PPO Hyperparameters

| Hyperparameter | Value |
| --- | --- |
| Normalize Advantage per Mini-Batch | True |
| Value Loss Coefficient | 1.0 |
| Clip Parameter | 0.2 |
| Use Clipped Value Loss | True |
| Desired KL | 0.01 |
| Entropy Coefficient | 0.01 |
| Discount Factor (Gamma) | 0.99 |
| GAE Lambda (Lam) | 0.95 |
| Max Gradient Norm | 1.0 |
| Learning Rate | 0.0003 |
| Number of Learning Epochs | 5 |
| Number of Mini-Batches | 16 |
| Schedule | Adaptive |
| Policy Class Name | ActorCritic |
| Activation Function | ELU |
| Actor Hidden Dimensions | [512, 512, 512] |
| Critic Hidden Dimensions | [512, 512, 512] |
| Initial Noise Std | 1.0 |
| Noise Std Type | Scalar |
| Number of Steps per Environment | 24 |
| Max Iterations | 2000 |
| Empirical Normalization | True |
| Number of Environments | 2048 |

## A.2 Simulation

We use Maniskill3 [62] for our simulations. Our hyperparameters are listed in Table 2. We situate our tasks in simulated scenes from the ReplicaCAD dataset [61]. Some of the objects in the tasks are from the RoboCasa project [47].

## A.3 VLM

We detail our query settings in Table 3.

# B Environment

## B.1 Observation Space

Table 4 details the components of our observation space. Importantly, our policy does not rely on privileged information such as contact forces during training, making the observation space more amenable to real-world deployment.

## B.2 Action Space

We use a residual action space on the wrist pose, and directly control the fingers. We normalized the action space to the range [-1, 1]. A "zero" action corresponds to following the reference trajectory precisely with an entirely open hand.

**Table 2:** Simulation and Control Settings

| Setting | Value |
|---|---|
| Action Exponential Average Gamma | 0.9 |
| Simulation Frequency | 120 Hz |
| Control Frequency | 60 Hz |
| Max Rigid Contact Count | $2048 \times 2048 \times 8$ |
| Max Rigid Patch Count | $2048 \times 2048 \times 2$ |
| Found Lost Pairs Capacity | $2^{27}$ |
| Gravity | [0, 0, -9.81] |
| Bounce Threshold | 2.0 |
| Solver Position Iterations | 8 |
| Solver Velocity Iterations | 0 |
| Default Dynamic Friction | 1.0 |
| Default Static Friction | 1.0 |
| Restitution | 0 |
| Finger Static Friction | 2.0 |
| Dummy Joint Stiffness | 2000 |
| Dummy Joint Damping | 100 |
| Dummy Joint Force Limits | 1000 |
| Finger Joint Stiffness | 10 |
| Finger Joint Damping | 0.3 |
| Finger Joint Force Limit | 10 |
| Controller Type | PD Joint Targets |

**Table 3:** VLM Configuration

| Hyperparameter | Value |
|---|---|
| Image Size | $800 \times 800$ |
| Trajectory Query Thinking Budget | 1000 |
| Keypoint Query Temperature | 0.5 |
| Trajectory Query Code Execution | Enabled |

We only control the fingers individually for the scissors and pliers tasks since we did not see any benefit for the other tasks. For the scissors and pliers task we have one action for every finger (instead of every joint). This makes hand action space four dimensional for the allegro hand. For all other tasks we control all fingers with one action, only opening or closing the entire hand.

### B.3 Termination Thresholds

We detail our initial termination thresholds per task in Table 5. The initial thresholds are linearly reduced to half of their initial value over the course of training.

## C Hardware Experiment Details

### C.1 Inference-Time Pipeline Details

At inference-time, we run our policy in the real-world as follows:

1. Capture an RGB-D image of the scene.
2. Query the VLM for 2D keypoints, given the RGB image and natural language command.
3. Backproject the 2D keypoints into 3D keypoints using the depth camera and camera intrinsics.
4. Transform the 3D keypoints from camera frame to world frame using camera extrinsics.
5. Query the VLM for a wrist pose trajectory and keypoint trajectories, given the initial wrist pose and 3D keypoints.

**Table 4:** Observation Space Configuration

| Observation Type | Dimension |
|---|---|
| Joint Position (Dummy Joints + Fingers) | 22 |
| Joint Velocity (Dummy Joints + Fingers) | 22 |
| Exponential Average Action | 22 |
| Finger Poses | $4 \times 7 = 28$ |
| Initial Keypoint Positions | $3 \times k$ |
| Current Keypoint Positions | $3 \times k$ |
| Planned Future Keypoint Positions | $3 \times 15 \times k$ |
| Current Wrist Pose | 6 |
| Planned Future Wrist pose | $6 \times 15 = 30$ |
| **Total** | $130 + 6 \times k + 3 \times 15 \times k$ |

**Table 5:** Initial Termination Thresholds for Manipulation Tasks

| Object | Threshold (cm) |
|---|---|
| Apple | 10 |
| Bottle | 10 |
| Hammer | 8 |
| Drawer | 15 |
| Sponge | 8 |
| Plier | 5 |
| Scissors | 3 |
| Fridge | 20 |

6. Run FoundationPose [70] to track the objects, which allows us to track their associated keypoints (we assume the keypoint does not move relative to the object frame)

7. Run the low-level policy, given the base wrist pose trajectory and tracked keypoints.

### C.2 Mapping Wrist Actions to Arm Joints

In simulation-only experiments, we control a *floating* (non-physical) hand whose wrist pose can be commanded directly. To train a policy in simulation that can be executed on a real robot, the wrist is attached to a 7-DoF KUKA LBR iiwa 14 arm, so the residual wrist-pose action produced by the policy must be converted into incremental arm joint commands. We perform this conversion with damped–least–squares inverse kinematics (DLS-IK).

Let $J \in \mathbb{R}^{6 \times N_{\text{arm}}}$ be the analytical Jacobian of the arm ($N_{\text{arm}} = 7$ in our setup), evaluated at the current joint configuration $\boldsymbol{\theta} \in \mathbb{R}^{N_{\text{arm}}}$, and let $\mathbf{e} \in \mathbb{R}^6$ be the 6-D spatial error twist (concatenated position and orientation error) between the current wrist pose and the target wrist pose. The arm joint update $\Delta \boldsymbol{\theta} \in \mathbb{R}^{N_{\text{arm}}}$ is computed as:

$$\Delta \boldsymbol{\theta} = J^\top \big( J J^\top + \lambda^2 I_6 \big)^{-1} \mathbf{e}, \tag{5}$$

where $\lambda = 0.5$ is a constant damping factor. Equation (5) implements the damped pseudoinverse $J_\lambda^\dagger = J^\top (J J^\top + \lambda^2 I_6)^{-1}$, yielding the minimum-norm solution to $J \Delta \boldsymbol{\theta} = \mathbf{e}$ while regularising the update near kinematic singularities. Lastly, we compute a joint position target $\boldsymbol{\theta}^{\text{target}} = \boldsymbol{\theta} + \Delta \boldsymbol{\theta}$, clamp this to stay within the joint limits, and then send this as the target to a low-level joint-position PD controller running at 200 Hz.

By default, the target wrist pose is specified by the wrist pose trajectory generated by the VLM. The policy outputs a residual wrist pose action that modifies this target, allowing fine-grained corrections. The resulting target is then used to compute the spatial error $\mathbf{e}$. By construction, if the residual wrist pose action is $\mathbf{0}$, the error $\mathbf{e}$ corresponds exactly to the difference between the current wrist pose and the original VLM-generated trajectory, so the arm will simply follow the given wrist pose trajectory.

## C.3 Tasks

We evaluate our system on three real-world manipulation tasks:

- **Slide Box to Bottle**: The goal is to push the box to the bottle. The box starts from a face-down orientation approximately 35cm away from the bottle. A trial is considered successful if the box makes contact with the bottle.

- **Place Bottle onto Plate**: The goal is to grasp the bottle and place it onto a plate. The bottle starts from an upright orientation approximately 42cm away from the plate. The task is considered successful if the bottle is lifted and makes contact with the top surface of the plate.

- **Hammer Three Times**: The goal is to grasp a hammer by its handle and strike a fixed surface three times in succession. The hammer starts in a resting position on the table within the robot's reachable workspace. A trial is considered successful if three distinct strikes are executed with observable contact between the hammer head and the table surface. This task requires precise control of contact dynamics and pose stabilization, making it substantially more challenging than pushing or pick-and-place tasks.

For each task, we run 20 trials across 4 VLM-generated trajectories. The procedure is as follows: We first initialize the objects in random positions with the same range as used in simulation training. Next, we capture an RGB-D image of the scene, and then query the VLM to generate a wrist trajectory and keypoint trajectories based on the image and a natural language instruction. Each generated trajectory is tested in 5 repeated trials, resetting the objects to similar initial poses before each attempt. This process is repeated 4 times with new randomized object positions and new trajectory queries, resulting in 20 total trials per task.

## C.4 Domain Randomization

We improve the policy's ability to transfer to the real world in a zero-shot setting through domain randomization. This enables robustness to physical parameters that are unknown, noisy, or inaccurately modeled in the real environment. Specifically, we apply the following randomizations during training:

- **Joint stiffness and damping** are multiplied from their default values by a factor sampled from a uniform distribution: $\mathcal{U}(0.3,\ 3.0)$. These parameters are sampled once at the start of training (independently for each parallel environment) and remain fixed throughout training.

- **Observation noise** is added to each of the robot proprioception observations, sampled independently from a normal distribution: $\mathcal{N}(0,\ 0.05^2)$. This is uncorrelated noise that is sampled at every control timestep.

- **Action noise** is added to exponential average action, sampled from: $\mathcal{N}(0,\ 0.05^2)$. This is uncorrelated noise that is sampled at every control timestep.

## C.5 Additional Adjustments

- The observation space is nearly identical to that described in Table 4, except that the dummy joints used to control the floating-hand wrist pose are replaced with the arm's actual joints (for joint positions, velocities, and exponentially averaged actions).

- We increase the exponential smoothing factor for the action average to $\gamma = 0.98$ to produce smoother motions and reduce jitter in the executed actions.

- We adjusted the trajectories to be twice as long for real-world experiments to effectively slow the robot motion down. We found that higher-speed motions typically resulted in less reliable policies, as this likely worsened the sim-to-real gap.

- To prevent significant collisions between the hand and the table, we clamp the $z$-coordinate of the target wrist pose to remain above the table height.

## C.6   Digital Twin Construction

Our digital twin simulation environment consisted of a robot, table, and two objects per task. The robot URDF and physics parameters were acquired by standard open-source repositories. We measured the dimensions of the table and its position relative to the robot with a measuring tape, which took about 5 minutes. The objects were scanned using an off-the-shelf 3D LiDAR scanning app called Kiri Engine, which took about 3 minutes per object.

## C.7   Kinematic Reachability

Object positions are initialized to ensure kinematic feasibility within the robot's workspace. Although VLMs are not explicitly aware of robot kinematics, our few-shot refinement process biases the model toward feasible trajectories by conditioning on successful prior samples. Furthermore, we introduce substantial noise during plan sampling, encouraging diversity of action proposals such that a significant number of the sampled plans during RL training are kinematically viable.

## C.8   Additional Qualitative Analysis

- We found that VLM keypoint detection worked significantly better on real world images, as they are more likely to be in-distribution than simulation images.

- As the low-level policy operates in a closed-loop fashion, we find it to be robust to dynamics differences between simulation and reality.

- The low-level RL policy appeared to optimize the task objective (moving the object keypoint along the generated keypoint trajectory) very well. For example, on the **Slide Box to Bottle** task, when the predicted box keypoint was on the bottom side of the box, the policy would not only push the box to the bottle, but rotate the box so that the bottom side of the box would be as close as possible to the bottle. On the **Place Bottle onto Plate** task, when the predicted bottle keypoint was on the upper half of the bottle, the policy would often place the bottle on its side so that the keypoint would be as close to the plate as possible.

- The most common failure mode came from the keypoint tracking errors. While the initial predicted keypoints were accurate, our pose tracker was only reliable when the object was completely unoccluded. The pose predictions got worse when the object was occluded and occasionally got very bad when highly occluded, which degraded policy performance.

- We performed preliminary experiments testing our policy on unseen objects with different but similar geometry (e.g., replacing the bottle with a mustard bottle or tall cup, replacing the plate with a different sized plate). The policy still worked reasonably well on these unseen objects due to the VLM's common-sense understanding to select good keypoints and the RL policy's state-based observations.

## D   Open Vision-Language Models

While our main experiments use Gemini 2.5 Flash Thinking, we additionally evaluate whether open-source VLMs can serve as effective alternatives within our framework. Specifically, we test Molmo [10], a recent open VLM designed for general multimodal understanding.

We evaluate Molmo on three representative tasks from our simulated benchmark: drawer opening, hammer, and pliers. For each task, both Gemini and Molmo are queried using identical prompts to generate 2D keypoints and trajectories. A keypoint prediction is counted as correct if it corresponds to a semantically meaningful and geometrically relevant location on the object, and a trajectory is considered feasible if it produces a physically executable motion in simulation.

Molmo successfully identified accurate and semantically relevant keypoints across all evaluated tasks but consistently failed to produce physically plausible or coherent trajectories. This suggests that while open VLMs like Molmo demonstrate strong visual grounding and scene understanding, they currently lack the higher-level spatial reasoning needed for multi-step action sequencing.

In contrast, Gemini 2.5 Flash Thinking generated both correct keypoints and feasible trajectories in the majority of trials. This can be attributed to its broader training on complex multimodal reasoning

**Table 6:** Comparison between an open VLM (Molmo) and a proprietary general-purpose VLM (Gemini 2.5 Flash Thinking) on three simulated tasks.

| Model | Drawer | | Hammer | | Pliers | |
|---|---|---|---|---|---|---|
| | Keypoints (%) | Traj. (%) | Keypoints (%) | Traj. (%) | Keypoints (%) | Traj. (%) |
| Molmo | 100 (10/10) | 0 (0/10) | 100 (10/10) | 0 (0/10) | 100 (10/10) | 0 (0/10) |
| Gemini | 100 | 87 | 87 | 67 | 90 | 80 |

tasks, which supports stronger high-level planning capabilities, critical for generating structured motion scaffolds in our setting.

Further benchmarking across open VLMs could provide additional insights into model-specific strengths, but such exploration is considered orthogonal to the core contribution of this work, which focuses on the framework design rather than VLM selection.

# E    Additional Baselines and Ablations

## E.1    Reinforcement Learning (RL)

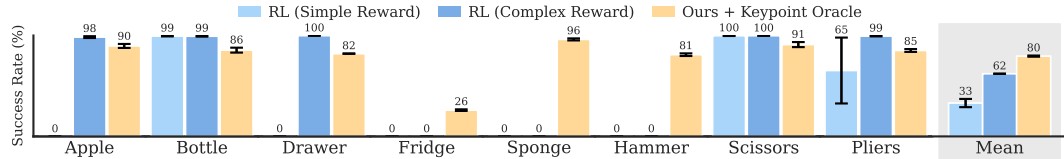

**Figure 9:** Results on the simulation task suite. Success rate (in %) is averaged across three seeds; uncertainty reflects the standard error. Our method performs comparably to the oracle with perfectly scripted plans.

We introduce two additional RL-based baselines: one with a simple reward function and another with a complex, task-specific reward.

**RL with Simple Reward.** In this baseline, a policy is trained from scratch using reinforcement learning (RL). The reward function mirrors that of our main method, combining contact and keypoint-distance-to-target terms, but uses *oracle keypoints* rather than those detected by the VLM. Importantly, this baseline does not use any trajectory guidance or demonstrations.

**RL with Complex Reward.**    This baseline employs a more detailed, hand-crafted, task-specific reward function that includes contact, keypoint-distance-to-target, hand-distance-to-object, and an additional success signal (assuming access to a success detector). This setup is intended to approximate the upper bound of performance achievable through extensive manual reward engineering. As with the previous baseline, no trajectory supervision or demonstrations are used.

To ensure a fair comparison, we evaluate it alongside our method under identical oracle keypoint conditions (i.e., *Ours + keypoint oracle*), while still relying on VLM-generated trajectories. Policies are trained using PPO. The results are summarized in Fig. 9.

## E.2    Imitation Learning (IL)

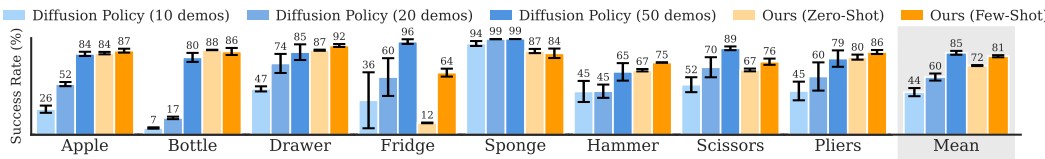

**Figure 10:** Results of imitation learning baselines on the simulation task suite. Success rate (in %) is averaged across three seeds, with uncertainty represented by the standard error. Our method performs comparably to Diffusion policy with 50 *perfect* demonstrations.

For imitation learning evaluation, we train a Diffusion Policy [8] model using successful rollouts from the oracle RL policy as demonstrations. Experiments are conducted with 10, 20, and 50 demonstrations. This represents a best-case scenario for imitation learning, as the demonstrations are *perfect*, originating from scripted trajectories and the oracle RL policy, rather than noisy human demonstrations. Success rates are reported with the mean standard error computed over three random seeds per task in Fig. 10.

### E.3 Noisy VLM Predictions

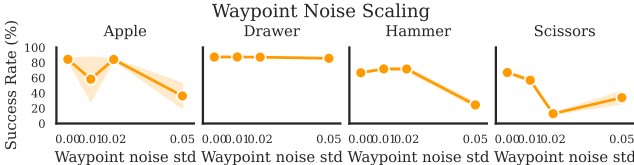

**Figure 11:** Effect of Gaussian noise on VLM predictions in the simulation task suite. Success rate (in %) is averaged across three seeds; uncertainty indicates the standard error.

The performance of our method depends on the accuracy of the VLM's outputs. Errors in keypoint detection or trajectory prediction can adversely affect downstream RL performance.

To quantify this sensitivity, we report results using oracle keypoint and trajectory predictions, representing an upper bound corresponding to perfect VLM outputs, in the main paper (see 7). Oracle predictions yield varying levels of performance improvement across tasks, and in some cases enable near-perfect success rates.

We additionally evaluate the opposite case by introducing artificial noise into the VLM outputs. Specifically, we add Gaussian noise $\mathcal{N}(0, \sigma^2)$ to the waypoints proposed by the VLM, thereby simulating planning errors. Noise is injected during both training and inference. Success rates are reported as the mean standard error computed over three random seeds per task in Fig. 11.

## F   Tasks

We provide brief descriptions of the eight simulated tasks we evaluated:

- **Move Apple:** An apple and a cutting board are placed on a kitchen counter. The keypoints are the apple and the cutting board. The agent's objective is to pick up the apple and place it on top of the cutting board.

- **Move Bottle:** A bottle is positioned on a kitchen counter next to a sink, either lying down or standing upright. The keypoints are the bottle and a target point on the counter across the sink. The goal is for the agent to pick up the bottle and place it upright on the opposite side of the sink.

- **Open Drawer:** A closed cupboard with multiple drawers is located in a living room. The handle of the top drawer serves as the sole keypoint. The objective is to open this drawer by at least 20 cm.

- **Open Fridge:** A closed refrigerator is situated in a kitchen. The handle of the fridge is the only keypoint. The agent's task is to open the fridge door by at least 60 degrees.

- **Hammer:** A hammer rests on a kitchen counter, with the head and handle defined as keypoints. The goal is for the agent to pick up the hammer and perform a hammering motion with at least three swings. A swing is defined as an upward and downward movement of at least 5 cm.

- **Wipe with Sponge:** A sponge is located on a kitchen counter, acting as the sole keypoint. The task is to perform a wiping motion on the counter, with success defined as moving the sponge at least 30 cm on the counter.

- **Close Scissors:** An open pair of scissors is situated on a kitchen counter, with the handles serving as keypoints. The goal is to close the scissors until the blades form an angle of less than 5 degrees.

- **Close Pliers:** An open pair of pliers is positioned on a kitchen counter, with the handles defined as keypoints. The objective is to close the pliers until the handles form an angle of less than 5 degrees.

# G  Compute Resources

Our training is performed on NVIDIA GPUs, ranging from A5000s to L40s. Depending on the specific task and hardware configuration, training durations vary between 1.5 and 6 hours. For real-world inference, we utilize two RTX 4090 GPUs.

# H  Prompt Examples

## H.1  Keypoint Elicitation Prompt

```
You are an expert in robot manipulation. Above is an image of the environment. Your
task is to {task_description} with a robot hand.
Identify a **minimal set of keypoints** needed to plan the motion. Avoid small,
sharp, or occluded areas like corners or edges. Choose large, stable regions visible
 from many angles.
Use **spatially descriptive, short names** (e.g., "left handle", "top surface") and
avoid ambiguous labels like "part 1". Only include points that are essential for
completing the task.
Reply with a JSON list of keypoint names.
```

## H.2  Move Apple

### H.2.1  Keypoint Prompt

```
Point to the apple and the cutting board in the image.
The answer should follow the json format: [{"name": "apple", "point": [...]}, {"name
": "cutting board", "point": [...]}]
The points are in [y, x] format normalized to 0-1000.
```

### H.2.2  Trajectory Prompt

```
Your are controlling a robot hand to pick up an apple and put it on a cutting board.
The coordinates are measured in meters.
The x axis is forward, the y axis is left and the z axis is up.
First move the robot hand towards the apple.
Then grasp the apple and lift it up.
Finally move the apple on the cutting board and put it down.
Describe a very realistic trajectory of exactly 20 waypoints.
Use code to generate the output.
The initial position of the apple is [0.00, 0.00, 0.00].
The initial position of the cutting board is [-0.01, -0.38, -0.05].
The initial position of the hand is [-0.07, -0.09, 0.26].
Use the following json format for the trajectory:
[{
"waypoint_num": 0,
"apple": {"x": float, "y": float, "z": float},
"cutting board": {"x": float, "y": float, "z": float},
"hand": {"x": float, "y": float, "z": float}
} ...]
**Only** print the json output. Do **not** print anything else with the code.
```

## H.3  Move Bottle

### H.3.1  Keypoint Prompt

```
Point to the water bottle on the kitchen counter, and pinpoint a point on the
kitchen counter to the right of the kitchen sink in the image.
The answer should follow the json format: [{"name": "bottle", "point": [...]}, {"
name": "point", "point": [...]}]
The points are in [y, x] format normalized to 0-1000.
```

### H.3.2 Trajectory Prompt

```
Your are controlling a robot hand to move a bottle to the target position called "
point" on the kitchen counter.
The coordinates are measured in meters.
The x axis is forward, the y axis is left and the z axis is up.
First move the robot hand towards the bottle.
Then grasp the bottle and lift it up.
Finally move the bottle to the target position called "point" and put it down.
Describe a very realistic trajectory of exactly 20 waypoints.
Use code to generate the output.
The initial position of the bottle is [0.00, 0.00, 0.00].
The initial position of the point is [-0.22, 0.80, -0.13].
The initial position of the hand is [0.25, -0.08, 0.20].
Use the following json format for the trajectory:
[\{
"waypoint_num": 0,
"bottle": {"x": float, "y": float, "z": float},
"point": {"x": float, "y": float, "z": float},
"hand": {"x": float, "y": float, "z": float}
\} ...]
**Only** print the json output. Do **not** print anything else with the code.
```

### H.4 Open Drawer

### H.4.1 Keypoint Prompt

```
Point to the handle of the top cabinet drawer in the image.
The answer should follow the json format: [{"name": "handle", "point": [...]}]
The points are in [y, x] format normalized to 0-1000.
```

### H.4.2 Trajectory Prompt

```
Your are controlling a robot hand to pull open a cabinet drawer.
The coordinates are measured in meters.
The x axis is forward, the y axis is left and the z axis is up.
First move the robot hand towards the handle of the drawer.
Then grasp the handle.
Finally pull the drawer open by at least 30cm.
Describe a very realistic trajectory of exactly 20 waypoints.
Use code to generate the output.
The initial position of the handle is [0.00, 0.00, 0.00].
The initial position of the hand is [0.32, -0.05, 0.12].
Use the following json format for the trajectory:
[{
"waypoint_num": 0,
"handle": {"x": float, "y": float, "z": float},
"hand": {"x": float, "y": float, "z": float}
} ...]
**Only** print the json output. Do **not** print anything else with the code.
```

### H.5 Open Fridge

#### H.5.1 Keypoint Prompt

```
Point to the top handle of the refrigerator door in the image.
The answer should follow the json format: [{"name": "handle", "point": [...]}]
The points are in [y, x] format normalized to 0-1000.
```

#### H.5.2 Trajectory Prompt

```
Your are controlling a robot hand to open a refrigerator door.
The coordinates are measured in meters.
The x axis is forward, the y axis is left and the z axis is up.
The refrigerator faces in x direction.
The y axis points to the right, and the z axis points up.
First figure out how large the door is.
Then describe how the x and y coordinates of the handle change as the door is opened
.
Now move the robot hand towards the handle.
Then grasp the handle.
Finally fully open the door.
Describe a very realistic trajectory of exactly 20 waypoints.
Use code to generate the output.
The initial position of the handle is [0.00, 0.00, 0.00].
The initial position of the hand is [0.50, 0.00, -0.22].
Use the following json format for the trajectory:
[{
"waypoint_num": 0,
"handle": {"x": float, "y": float, "z": float},
"hand": {"x": float, "y": float, "z": float}
} ...]
**Only** print the json output. Do **not** print anything else with the code.
```

### H.6 Hammer

#### H.6.1 Keypoint Prompt

```
Point to the brown handle and the metal head of the hammer in the image.
The answer should follow the json format: [{"name": "handle", "point": [...]}, {"
name": "head", "point": [...]}]
The points are in [y, x] format normalized to 0-1000.
```

#### H.6.2 Trajectory Prompt

```
Your are controlling a robot hand to make a hammering motion.
The coordinates are measured in meters.
The x axis is forward, the y axis is left and the z axis is up.
First move the robot hand towards the handle.
Then grasp the handle.
Finally hit on the kitchen counter 3 times.
Describe a very realistic trajectory of exactly 20 waypoints.
Use code to generate the output.
The initial position of the handle is [0.00, 0.00, 0.00].
The initial position of the head is [-0.02, -0.15, 0.03].
The initial position of the hand is [0.01, 0.06, 0.27].
Use the following json format for the trajectory:
[{
"waypoint_num": 0,
"handle": {"x": float, "y": float, "z": float},
"head": {"x": float, "y": float, "z": float},
"hand": {"x": float, "y": float, "z": float}
```

```
} ...]
**Only** print the json output. Do **not** print anything else with the code.
```

### H.7   Wipe with Sponge

#### H.7.1   Keypoint Prompt

```
Point to the green yellow sponge on the kitchen counter in the image.
The answer should follow the json format: [{"name": "sponge", "point": [...]}]
The points are in [y, x] format normalized to 0-1000.
```

#### H.7.2   Trajectory Prompt

```
Your are controlling a robot hand to wipe a kitchen counter with a sponge.
The coordinates are measured in meters.
The x axis is forward, the y axis is left and the z axis is up.
First move the robot hand towards the sponge.
Then grasp the sponge.
Finally wipe the kitchen counter with the sponge.
Describe a very realistic trajectory of exactly 20 waypoints.
Use code to generate the output.
The initial position of the sponge is [0.00, 0.00, 0.00].
The initial position of the hand is [0.26, 0.03, 0.29].
Use the following json format for the trajectory:
[{
"waypoint_num": 0,
"sponge": {"x": float, "y": float, "z": float},
"hand": {"x": float, "y": float, "z": float}
} ...]
**Only** print the json output. Do **not** print anything else with the code.
```

### H.8   Close Scissors

#### H.8.1   Keypoint Prompt

```
Point to the two loops of the scissors in the image.
The answer should follow the json format: [{"name": "loop 1", "point": [...]}, {"
name": "loop 2", "point": [...]}]
The points are in [y, x] format normalized to 0-1000.
```

#### H.8.2   Trajectory Prompt

```
Your are controlling a robot hand to close a pair of scissors.
The coordinates are measured in meters.
The x axis is forward, the y axis is left and the z axis is up.
First move the robot hand towards the scissors.
Then grasp the two loops and entirely close the scissors.
Describe a very realistic trajectory of exactly 20 waypoints.
Use code to generate the output.
The initial position of the loop 1 is [0.00, 0.00, 0.00].
The initial position of the loop 2 is [-0.07, 0.07, 0.01].
The initial position of the hand is [0.03, -0.06, 0.33].
Use the following json format for the trajectory:
[{
"waypoint_num": 0,
"loop 1": {"x": float, "y": float, "z": float},
"loop 2": {"x": float, "y": float, "z": float},
"hand": {"x": float, "y": float, "z": float}
} ...]
**Only** print the json output. Do **not** print anything else with the code.
```

### H.9 Close Pliers

#### H.9.1 Keypoint Prompt

```
Point to the left and right handles of the plier in the image.
The answer should follow the json format: [{"name": "handle left", "point": [...]},
{"name": "handle right", "point": [...]}]
The points are in [y, x] format normalized to 0-1000.
```

#### H.9.2 Trajectory Prompt

```
Your are controlling a robot hand to close a plier.
The coordinates are measured in meters.
The x axis is forward, the y axis is left and the z axis is up.
First move the robot hand towards the plier.
Then grasp the left and right handles and entirely close the plier.
Describe a very realistic trajectory of exactly 20 waypoints.
Use code to generate the output.
The initial position of the handle left is [0.00, 0.00, 0.00].
The initial position of the handle right is [-0.05, 0.16, 0.00].
The initial position of the hand is [0.01, -0.08, 0.31].
Use the following json format for the trajectory:
[{
"waypoint_num": 0,
"handle left": {"x": float, "y": float, "z": float},
"handle right": {"x": float, "y": float, "z": float},
"hand": {"x": float, "y": float, "z": float}
} ...]
**Only** print the json output. Do **not** print anything else with the code.
```

### H.10 Place Bottle onto Plate

#### H.10.1 Keypoint Prompt

```
Point to the middle of the bottle and the plate on the table in the image.
The answer should follow the json format: [{"name": "bottle", "point": [...]}, {"
name": "plate", "point": [...]}]
The points are in [y, x] format normalized to 0-1000.
```

#### H.10.2 Trajectory Prompt

```
Your are controlling a robot hand to move a bottle onto a plate.
The coordinates are measured in meters.
The x axis is forward, the y axis is left and the z axis is up.
First move the robot hand towards the bottle.
Then grasp the bottle and lift it up.
Then place the bottle on to the plate.
Describe a very realistic trajectory of exactly 20 waypoints.
Use code to generate the output.
The initial position of the bottle is [0.00, 0.00, 0.00].
The initial position of the plate is [0.30, 0.38, -0.15].
The initial position of the hand is [-0.15, -0.29, 0.09].
Use the following json format for the trajectory:
[{
"waypoint_num": 0,
"bottle": {"x": float, "y": float, "z": float},
"plate": {"x": float, "y": float, "z": float},
"hand": {"x": float, "y": float, "z": float}
} ...]
**Only** print the json output. Do **not** print anything else with the code.
```

### H.11 Slide Box to Bottle

#### H.11.1 Keypoint Prompt

```
Point to the box and the bottle on the table in the image.
The answer should follow the json format: [{"name": "box", "point": [...]}, {"name":
 "bottle", "point": [...]}]
The points are in [y, x] format normalized to 0-1000.
```

#### H.11.2 Trajectory Prompt

```
Your are controlling a robot hand to slide the box over the table to the bottle.
The coordinates are measured in meters.
The x axis is forward, the y axis is left and the z axis is up.
First move the robot hand towards the box.
Then slide the box over the table to the bottle.
Describe a very realistic trajectory of exactly 20 waypoints.
Use code to generate the output.
The initial position of the box is [0.00, 0.00, 0.00].
The initial position of the bottle is [0.17, 0.23, 0.08].
The initial position of the hand is [-0.22, -0.29, 0.19].
Use the following json format for the trajectory:
[{
"waypoint_num": 0,
"box": {"x": float, "y": float, "z": float},
"bottle": {"x": float, "y": float, "z": float},
"hand": {"x": float, "y": float, "z": float}
} ...]
**Only** print the json output. Do **not** print anything else with the code.
```

### H.12 Hammer Three Times

#### H.12.1 Keypoint Prompt

```
Point to the handle and the head of the hammer in the image.
The answer should follow the json format: [{"name": "handle", "point": [...]}, {"
name": "head", "point": [...]}]
The points are in [y, x] format normalized to 0-1000.
```

#### H.12.2 Trajectory Prompt

```
Your are controlling a robot hand to make a hammering motion.
The coordinates are measured in meters.
The x axis is forward, the y axis is left and the z axis is up.
First move the robot hand towards the hammer handle.
Then grasp the hammer handle.
Finally hit on the kitchen counter 3 times.
Describe a very realistic trajectory of exactly 20 waypoints.
Use code to generate the output.
The initial position of the handle is [0.00, 0.00, 0.00].
The initial position of the head is [0.19, 0.13, -0.04].
The initial position of the hand is [0.12, 0.01, 0.12].
Use the following json format for the trajectory:
[{
"waypoint_num": 0,
"handle": {"x": float, "y": float, "z": float},
"head": {"x": float, "y": float, "z": float},
"hand": {"x": float, "y": float, "z": float}
} ...]
**Only** print the json output. Do **not** print anything else with the code.
```

