# OpenReview forum: "Scaffolding Dexterous Manipulation with Vision-Language Models"
_NeurIPS.cc/2025/Conference — NeurIPS 2025 poster_

### Official Review · Reviewer_b94o · 2025-06-18

**Clarity:** 3
**Significance:** 3
**Originality:** 3
**Rating:** 3
**Confidence:** 4

**Summary:**

This paper proposes a two-stage framework for general dexterous manipulation. The first stage uses a VLM to generate semantic keypoints and corresponding trajectories, while the second stage employs reinforcement learning to train a residual policy for low-level control. Both simulation and real-world experiments are conducted to validate the effectiveness of the method.

**Questions:**

The current method requires successful trajectories from the training set to enhance VLM planning. How can this method be extended to unseen scenes and objects?

**Ethical Concerns:**

["NO or VERY MINOR ethics concerns only"]

**Final Justification:**

While the author provides additional justification and experiments, I think the unrealistic motion shown in the video still exposes the limitation of the current method, and the experiment provided by the author also validates the concern. Therefore, I think the current method is not a very ready paper to show the potential of the current method.

**Limitations:**

yes

**Quality:**

2

**Strengths And Weaknesses:**

Strengths:
1. This paper focuses on an important problem—current general dexterous manipulation methods mostly rely on reference trajectory imitation. However, sourcing such trajectories can be challenging. This paper addresses the issue by leveraging the capabilities of VLM.

Weaknesses:
1. The proposed keypoints and trajectories may not be fine-grained, yet the current method encourages the agent to imitate them. For example, in the hammer case, it actually grasps the head, causing clipping. Such a policy would likely to fail in real-world scenarios.
2. The keypoints proposed by VLM contain only positional information without orientation. For instance, if the hammer lies flat on the table rather than standing upright, the policy would not learn to reorient it. This missing information may limit the generalization ability of VLM.
3. From the simulation results, although the policy achieves a high success rate, the physical realism for complex manipulation seems questionable. For example, when closing the pliers, the dexterous hand does not actually grasp the object, yet it still completes the task. Additionally, the real-world experiments do not include more complex manipulation, which raises concerns about the actual capability of VLM in handling complex dexterous manipulation tasks.

---

> ### Author Rebuttal · Authors · 2025-07-31
>
> We thank the reviewer for the constructive feedback. Below, we address each of the key points raised.
>
> # Coarseness of Scaffolds
>
> Note that scaffolds are only intended to be used as a reward signal and to guide exploration – the policy is free to learn whatever it wants on top of the reward signal and thus is not strictly “imitating”.  We acknowledge that our scaffold-based reward signals are coarser than those derived from human demonstrations or hand-crafted dense rewards. However, this is a deliberate design choice, balancing scalability and supervision quality. Human demonstrations are often costly, time-intensive, and difficult to scale – especially as new tasks require collecting entirely new datasets. Similarly, crafting dense, high-quality reward functions demands significant manual engineering and domain expertise.
>
> In contrast, our scaffolds provide lightweight, task-agnostic, semantically meaningful supervision with minimal human input. While inherently less precise, we show that these signals are sufficiently informative to guide learning across a wide variety of tasks.
>
> We see scaffolds as a practical middle ground: they are significantly more scalable than demonstrations or hand-tuned rewards, while still effective across diverse domains. Future work may explore hybrid approaches – e.g., combining scaffold-based rewards with sparse success signals or contact-based feedback – to enable finer control when needed.
>
> # Lack of Explicit Orientation Information
>
> Though each keypoint does not encode orientation, multiple keypoints together do implicitly capture orientation when at least three non-colinear keypoints are annotated – an approach that is feasible for many real-world objects. In this way, motion scaffolds can effectively be used for tasks requiring rotation.
>
> To demonstrate this capability, we modified the original bottle task: the bottle may now start either lying down or standing upright on one side of a sink and must be placed upright on the opposite side. This setup requires the policy to infer the initial pose and adjust its behavior accordingly. We report a success rate of 0.64 ± 0.1 across three seeds. By using multiple spatial keypoints together, the policy learns to rotate and upright a bottle. Unfortunately, due to the limitations of rebuttal this year we are unable to attach images or videos.
>
> | Method               | Success Rate (mean ± SE) |
> |----------------------|--------------------------|
> | Pre-recorded baseline | 0.09 ± 0.02              |
> | Ours (zero-shot)      | 0.64 ± 0.10              |
>
> We acknowledge that for smaller objects, where one or two keypoints may be insufficient to disambiguate orientation, this becomes more challenging. In such cases, richer scaffold representations – potentially powered by advances in spatially grounded vision-language models (VLMs) – represent a promising direction to improve expressiveness and generality.
>
> # Physical Realism and Real-World Complexity
>
> We appreciate the reviewer’s concern regarding the complexity and physical realism of our tasks. To address this, we have extended our real-world evaluation to include a significantly more demanding **hammering** task, in which the robot must grasp a hammer by the handle and strike a table three times. This task demands substantially more precision and physical realism than pick-and-place or pushing tasks. Due to NeurIPS guidelines, we are unable to provide additional visual materials (e.g., images or videos) at this stage.
>
> Despite the increased difficulty, our system achieves a 60% success rate over 20 rollouts. Importantly, most failure cases arise not from deficiencies in the learning framework or simulation fidelity, but from hardware-related constraints, such as the mechanical limitations of the Allegro Hand [LEAP Hand, Shaw et al RSS 23] and noise in depth sensing [Learning to estimate 3d states of deformable objects from single-frame occluded point clouds, Lv et al 23].
>
> We acknowledge that sim-to-real transfer remains an open and well-recognized challenge, particularly for tasks requiring high-precision contact and control [ObjDex Chen et al. 24, RoboDexVLM Liu et al. 25, Solving a Rubrik’s Cube, OpenAI et al. 19]. As part of ongoing work, we plan to investigate improvements in simulation realism, multi-view visual perception, and sensor fusion – directions we believe are essential for narrowing the sim-to-real gap.
>
> # Generalization to Unseen Scenes and Objects
>
> We thank the reviewer for highlighting the importance of generalization. Our approach is designed to promote generalization by leveraging VLM-generated scaffolds and a policy conditioned on keypoint observations and proprioception, rather than raw visual input. This decouples the learning process from pixel-level variability and allows for greater robustness.
>
> Empirically, we observe strong zero-shot generalization in real-world settings. For example, in the bottle pick-and-place task, we successfully replaced the bottle with cups and mugs, maintaining high success rates by modifying only a single word in the prompt. Additionally, the system demonstrates robustness to changes in lighting conditions, background clutter, and instruction phrasing, with no significant degradation in performance across these variations.
>
> That said, we acknowledge that generalization to entirely new task categories or structurally different objects remains a significant challenge. In practice, doing so has required training policies across an extremely broad set of objects [DextrAH, Singh et al 24], or learning object agnostic representations. We see this as an important and exciting direction for future work in combination with our proposed motion scaffolds.

---

> > ### Author Response · Authors · 2025-08-05
> >
> > Dear reviewer,
> >
> > We would like to follow up to see if the response addresses your concerns or if you have any further questions. We would really appreciate the opportunity to discuss this further if our response has not already addressed your concerns. Thank you again!

---

> > ### Comment · Reviewer_b94o · 2025-08-07
> > **Respond to rebuttal**
> >
> > 1. Coarseness of Scaffolds: The reason I doubt the coarseness of scaffolds is that most of the hard task videos shown on the website do not show realistic motion, which makes me doubt whether such coarse scaffolds can serve as general guidance for diverse tasks.
> > 2. Lack of Explicit Orientation Information: I think this experiment exactly shows the limitation of the current method, by a large decrease in success rate on the harder task.
> > 3. Physical Realism and Real-World Complexity: To be honest, I am quite wondering how such unrealistic policy in sim can achieve 60% success rate in the real world. What changes did the author make, if not how such a policy could achieve such high performance given the demo in the sim actually collide with the hammer?

---

### Official Review · Reviewer_5Gyk · 2025-06-29

**Clarity:** 4
**Significance:** 2
**Originality:** 2
**Rating:** 4
**Confidence:** 4

**Summary:**

In this work,  the authors present a framework for dexterous manipulation which leverages: (i) a high-level VLM policy that infers (2D) keypoints in an image, and provides a (3D) trajectory for those keypoints and the hand's wrist that leads to the task solution, (ii) a low-level residual RL policy, which can correct the wrist trajectory to follow and control the hand's joints. By leveraging a detailed prompt structure and the pre-trained knowledge of a large VLM (Gemini 2.5 Flash Thinking), the framework achieves near-oracle performance in simulation and can transfer to real settings (after training on a digital twin in simulation).

**Questions:**

* What are the technical novelties introduced in the work, apart from the overall (novel) framework?
* What happens if we reduce the amount of "human supervision" in the prompts?
* What are some ways that the prompt structure could be made more general/modular? E.g. the user only inputs the task as "move the bottle to the right"
* Given that most of the evaluation is performed in simulation, demonstrations should be easy to produce (e.g. using the oracle). How does the approach compare to imitation learning approaches? E.g. how many demos do we need to reach the framework's performance?
* Given that most of the evaluation is performed in simulation, it should be possible to use SOTA RL techniques to solve the given tasks. How many timesteps/hours are needed to match the performance of the framework proposed?
* The authors claim their simple reward function is beneficial. This could be a useful novelty of the work. Could the authors compare to other reward designs to assess this?

**Ethical Concerns:**

["NO or VERY MINOR ethics concerns only"]

**Final Justification:**

I think the paper has plenty of value from an engineering perspective. The idea presented is novel and enables a framework that is flexible and performant. During the rebuttal, the authors have also improved the generalizability of the approach and have provided a more extensive empirical evaluation, generally improving the quality of the work.

From a machine learning perspective, the novelties introduced in the work are minor and mostly adapt previous ideas to work together seamlessly. Nonetheless, I am leaning towards acceptance of the paper, as I believe part of the NeurIPS community (those working on robotics) may find the work valuable.

**Limitations:**

The authors should discuss how their framework lacks generality and discuss how to generalize their prompting or get rid of it.

**Paper Formatting Concerns:**

No paper formatting concerns.

**Quality:**

3

**Strengths And Weaknesses:**

## Strengths
* **Design and engineering**: the framework presented is modular, e.g. the VLM can be upgraded with a newer version, and the components seem to fit well together, nicely allowing to leverage the advantages of each module, e.g. the pre-trained knowledge of the VLM or the residual RL policy for correcting coarse trajectories. The deployment on a real-world platform is also remarkable.
* **Presentation**: the work is well presented, with high-quality and helpful figures and great clarity.
## Weaknesses
* **Limited novelty**: the overall framework is, to the best of my knowledge, novel. However, the novelties introduced in the single components seem to be quite limited.
* **Limited generality**:  looking at the structure of the prompt provided in Appendix F, the framework seem to require task-dependent and environment-dependent prompts. Example keypoint prompt: "Point to the water bottle on the kitchen counter, and pinpoint a point on the kitchen counter to the right of the kitchen sink in the image.". This prompt contain specific information about the structure of the environment, the position of the bottle and detailed instructions on where to place the bottle. A similar limitation applies to the trajectory prompts. Thus, while the method requires no demonstration, it seems to require some form of human annotation to solve the task.
* **Limited comparison**: it's hard to assess the system if no baselines are provided. I understand the problem settings is somehow unique. However, there are several ways the authors could expand their comparison of the method. (see Questions)

---

> ### Author Rebuttal · Authors · 2025-07-31
>
> We thank the reviewer for their thoughtful and constructive feedback. Below, we address the key points regarding the novelty of our components, prompt design, and experimental baselines.
> # Novelty of Individual Components
>
> We respectfully disagree with the assessment that the individual components of our framework lack novelty. While our primary contribution lies in the system as a whole – which successfully enables real-world task execution, a recognized contribution in itself – this system is built on several novel and impactful ideas.
>
> To the best of our knowledge, no prior work leverages **VLMs to generate keypoint trajectories for guiding reinforcement learning**, nor uses **keypoint sequences as a scaffold for both reward shaping and exploration in RL**. Prior works have instead only used keypoints for rewards [IKER Patel et al 25] or used demonstrations for residual RL and tracking [ObjDex Chen et al 24]. Our work represents a significant departure from conventional approaches, which typically rely on these more expensive sources of supervision like motion capture, demonstrations, or scripted heuristics.
>
> We instead use structured, semantically meaningful guidance derived directly from natural language prompts to a VLM. This eliminates the need for demonstrations or complex setup procedures, and substantially lowers the cost and complexity of deploying RL in real-world manipulation tasks.
>
> # Prompt Specificity
>
> We appreciate the concern about the risk of prompt specificity introducing unintended task knowledge. In response, we revised our approach to use a **single, concise natural language instruction per task** (e.g., “grasp the apple and move it onto the cutting board” or “grasp the refrigerator handle and fully open the top refrigerator door”).
>
> To do this, we use a standardized preamble to elicit task-relevant keypoints from the VLM:
>
> *"You are an expert in robot manipulation. Above is an image of the environment. Your task is to {task_description} with a robot hand.
> Identify a minimal set of keypoints needed to plan the motion. Avoid small, sharp, or occluded areas like corners or edges. Choose large, stable regions visible from many angles.
> Use spatially descriptive names (e.g., "left handle", "top surface") and avoid ambiguous labels like "part 1". Only include points that are essential for completing the task.
> Reply with a JSON list of keypoint names."*
>
> The keypoints generated by the VLM using this prompt are employed both for keypoint detection and for motion trajectory planning. We find that the chosen keypoints closely align with human intuitions about salient features necessary for task completion.
>
> To assess the impact of the new prompting strategy, we evaluated performance across three manipulation tasks and report the mean success rate ± standard error over three seeds.
>
> | Prompt      | Apple       | Fridge      | Pliers      |
> |-------------|-------------|-------------|-------------|
> | Old Prompts | 0.84 ± 0.01 | 0.12 ± 0.00 | 0.80 ± 0.03 |
> | New Prompts | 0.88 ± 0.01 | 0.15 ± 0.01 | 0.81 ± 0.02 |
>
> These results indicate that simplifying and standardizing the prompts preserves task performance while reducing the risk of embedding task-specific priors, thereby improving the generality of our approach.
>
> # Baselines
>
> To strengthen the empirical evaluation, we have incorporated four additional baselines. We report success rates as the mean +- standard error computed over three seeds for each task.
>
> ## Iterative Keypoint Rewards (IKER)
>
> We implement an IKER-style baseline in which a vision-language model (VLM) generates code for reward function parameters, using the exact prompts and methodology from the original paper. However, as IKER assumes a fixed set of keypoints, we use the same VLM-identified keypoints as in our method for parity.
>
> | Method           | Apple       | Bottle      | Drawer      | Fridge      | Hammer      | Pliers      | Scissors    | Sponge      |
> |------------------|-------------|-------------|-------------|-------------|-------------|-------------|-------------|-------------|
> | Iker (zero-shot) | 0.45 ± 0.14 | 0.63 ± 0.08 | 0.23 ± 0.01 | 0.19 ± 0.04 | 0.53 ± 0.03 | 0.19 ± 0.11 | 0.27 ± 0.03 | 0.06 ± 0.02 |
> | Ours (zero-shot) | 0.84 ± 0.01 | 0.88 ± 0.00 | 0.87 ± 0.01 | 0.12 ± 0.00 | 0.67 ± 0.01 | 0.80 ± 0.03 | 0.67 ± 0.01 | 0.87 ± 0.02 |
>
> ## RL From Scratch
>
> We introduce an additional baseline in which a policy is trained from scratch using RL. The reward function mirrors that of our method – combining contacts and keypoint-distance-to-target – but relies on oracle keypoints rather than those detected by a VLM. Importantly, this baseline does not use any trajectory guidance or demonstrations. To ensure a fair comparison, we evaluate it alongside our method using the same oracle keypoints (i.e., Ours + keypoint oracle), but retaining VLM-generated trajectories. Policies are trained using PPO for the same number of timesteps as our method.
>
> | Method                      | Apple       | Bottle      | Drawer      | Fridge      | Hammer      | Pliers      | Scissors    | Sponge      |
> |-----------------------------|-------------|-------------|-------------|-------------|-------------|-------------|-------------|-------------|
> | Simple RL + keypoint oracle | 0.00 ± 0.00 | 0.99 ± 0.00 | 0.00 ± 0.00 | 0.00 ± 0.00 | 0.00 ± 0.00 | 0.65 ± 0.33 | 1.00 ± 0.00 | 0.00 ± 0.00 |
> | Ours + keypoint oracle      | 0.90 ± 0.02 | 0.85 ± 0.02 | 0.82 ± 0.00 | 0.26 ± 0.01 | 0.81 ± 0.01 | 0.85 ± 0.01 | 0.91 ± 0.02 | 0.96 ± 0.01 |
>
> ## Imitation Learning (IL)
>
> To evaluate imitation learning, we train Diffusion Policy using successful rollouts of our oracle RL policy as demonstrations. We conduct experiments using 10, 20, and 50 demonstrations. This represents the best-case performance for imitation learning: the demonstrations are perfect, resulting from scripted trajectories and an RL policy, unlike standard human demonstrations.
>
> | Method                      | Apple       | Bottle      | Drawer      | Fridge      | Hammer      | Pliers      | Scissors    | Sponge      |
> |-----------------------------|-------------|-------------|-------------|-------------|-------------|-------------|-------------|-------------|
> | Diffusion Policy (10 demos) | 0.26 ± 0.04 | 0.07 ± 0.01 | 0.47 ± 0.03 | 0.36 ± 0.29 | 0.45 ± 0.11 | 0.45 ± 0.10 | 0.52 ± 0.08 | 0.94 ± 0.03 |
> | Diffusion Policy (20 demos) | 0.52 ± 0.02 | 0.17 ± 0.01 | 0.74 ± 0.10 | 0.60 ± 0.20 | 0.45 ± 0.07 | 0.60 ± 0.15 | 0.70 ± 0.10 | 0.99 ± 0.00 |
> | Diffusion Policy (50 demos) | 0.84 ± 0.02 | 0.80 ± 0.05 | 0.85 ± 0.08 | 0.96 ± 0.02 | 0.65 ± 0.09 | 0.79 ± 0.08 | 0.89 ± 0.02 | 0.99 ± 0.00 |
> | Ours (few-shot)             | 0.87 ± 0.02 | 0.86 ± 0.04 | 0.92 ± 0.01 | 0.64 ± 0.04 | 0.75 ± 0.00 | 0.86 ± 0.02 | 0.76 ± 0.03 | 0.84 ± 0.05 |
> | Ours (zero-shot)            | 0.84 ± 0.01 | 0.88 ± 0.00 | 0.87 ± 0.01 | 0.12 ± 0.00 | 0.67 ± 0.01 | 0.80 ± 0.03 | 0.67 ± 0.01 | 0.87 ± 0.02 |
>
> # Comparison to Other Reward Designs
>
> The reviewer asked about comparisons to other, more complicated reward designs. To shed light on the effectiveness of our reward design, we additionally compare RL from scratch using more detailed, task specific, hand-crafted reward functions. This reward includes contact, keypoint-distance-to-target, hand-distance-to-object, and an additional ground truth success signal, assuming access to a success detector. This baseline is intended to reflect the upper bound of performance achievable through extensive manual reward engineering. As with the previous baseline, it does not utilize any trajectory supervision or demonstrations. To ensure a fair comparison, we evaluate it alongside our method using the same oracle keypoints (Ours + keypoint oracle). Policies are trained using PPO.
>
> | Method                       | Apple       | Bottle      | Drawer      | Fridge      | Hammer      | Pliers      | Scissors    | Sponge      |
> |------------------------------|-------------|-------------|-------------|-------------|-------------|-------------|-------------|-------------|
> | Complex RL + keypoint oracle | 0.98 ± 0.01 | 0.99 ± 0.00 | 1.00 ± 0.00 | 0.00 ± 0.00 | 0.00 ± 0.00 | 0.99 ± 0.00 | 1.00 ± 0.00 | 0.00 ± 0.00 |
> | Ours + keypoint oracle       | 0.90 ± 0.02 | 0.85 ± 0.02 | 0.82 ± 0.00 | 0.26 ± 0.01 | 0.81 ± 0.01 | 0.85 ± 0.01 | 0.91 ± 0.02 | 0.96 ± 0.01 |

---

> ### Comment · Reviewer_5Gyk · 2025-08-03
>
> I thank the authors for their rebuttal comments.
>
> As stated in the original review, I recognize that the overall framework is novel. With this, I am referring to the idea of using a VLM to generate keypoint trajectories and then adopting a RL-based residual policy for achieving such keypoints. However, it still remains unclear what are the novelties compared to the literature, in terms of machine learning techniques advancement. The VLM is a publicly available pre-trained model. The main difference, compared to other approaches, resides in the prompt. The RL policy is trained to track the keypoints provided by the VLM, while keeping contact with objects. In the literature, keypoint tracking is generally used for imitating mocap data [1,2], while contact rewards are more common in manipulation. Residual RL has also been investigated in the past [3].
>
> I hope the above clarifies why my general feeling is that no component of the framework introduces particular technical novelties. Nonetheless, I still believe that the idea proposed by the authors is valuable, interesting from an engineering perspective, and it makes for a flexible and performant approach.
>
> Other than this, I think the authors have fairly addressed my other concerns:
> * the more general prompt improves the applicability of the approach to new scenarios, without compromising performance
> * the experiments presented in the rebuttal improve the empirical validation of the approach by comparing with several baselines
>
> I will increase my rating of the paper accordingly.
>
> [1] Learning human behaviors from motion capture by adversarial imitation, Merel et al
>
> [2] CoMic: Complementary Task Learning & Mimicry for Reusable Skills, Hasenclever et al
>
> [3] Residual Reinforcement Learning for Robot Control, Johannink et al

---

### Official Review · Reviewer_5ndj · 2025-07-02

**Clarity:** 3
**Significance:** 3
**Originality:** 3
**Rating:** 4
**Confidence:** 3

**Summary:**

The paper tackles the long-standing challenge of training dexterous robotic hands without labor-intensive demonstrations or painstakingly engineered reward functions. The authors observe that the fine-grained details in traditional reference trajectories are largely superfluous, because reinforcement learning (RL) ultimately fine-tunes the motion anyway. Their key insight is to replace those trajectories with coarse spatial “scaffolds” automatically produced by a general-purpose vision-language model (VLM).

**Questions:**

Refer to the weakness section.

**Ethical Concerns:**

["NO or VERY MINOR ethics concerns only"]

**Final Justification:**

I would like to maintain my rating based on overall novelty of the paper and rebuttal made.

**Limitations:**

Yes

**Paper Formatting Concerns:**

Nill

**Quality:**

3

**Strengths And Weaknesses:**

Strength:
The framework sidesteps both manual reward engineering and demonstration collection by using VLM-generated 3-D trajectories as supervisory “scaffolds,” greatly reducing human effort and boosting scalability.

Re-querying the VLM to randomize keypoints and trajectories during training exposes the policy to diverse initial conditions, leading to strong generalization to unseen states and motion plans at test time.

Policies trained entirely in simulation transfer to a physical dexterous hand with robust performance, demonstrating practical feasibility without any human demonstrations or task-specific rewards.

Weakness:
-Have the author use robot specific VLM that are fine-tuned for spatial affordance prediction such as RoboPoint or even Molmo? Instead of Gemini.

-The method assumes the VLM can reliably pick out task-relevant key-points and generate a plausible 3-D scaffold. If key-point detection or trajectory synthesis is off (e.g., due to clutter, novel objects, bad lighting), the residual RL layer starts from a poor initialization and may converge to sub-optimal or unsafe motions. However, there isn't ablations that inject systematic VLM errors, nor does it quantify how much deviation the low-level controller can tolerate before success drops.

-The key-point/waypoint representation ignores force/torque constraints, finger–object contact states, and within-hand re-grasping. This abstraction works for the hammer, pliers, and bottle tasks shown, but tasks that hinge on precise in-hand posture changes (e.g., turning a screw two full revolutions) may require richer supervision than the scaffold can provide.

---

> ### Author Rebuttal · Authors · 2025-07-31
>
> We thank the reviewers for their constructive feedback. Below, we respond to the key points raised.
>
> # Use of Robot-Specific Vision-Language Models (VLMs)
>
> We appreciate the suggestion to explore robot-specific VLMs. As part of our investigation, we evaluated Molmo as an alternative to general-purpose models. We have tried detecting keypoints and generating scaffolds for 3 tasks. The table below shows the percentage of correct keypoints and feasible trajectories.
>
> |        | Drawer            |                   | Hammer            |                   | Pliers            |                   |
> |--------|-------------------|-------------------|-------------------|-------------------|-------------------|-------------------|
> |        | Correct Keypoints | Feasible Trajectories | Correct Keypoints | Feasible Trajectories | Correct Keypoints | Feasible Trajectories |
> | Molmo  | 100% (10/10)      | 0% (0/10)          | 100% (10/10)      | 0% (0/10)          | 100% (10/10)      | 0% (0/10)          |
> | Gemini | 100%              | 87%                | 87%               | 67%                | 90%               | 80%                |
>
> While Molmo reliably identified keypoints, it consistently failed to generate coherent plans and exhibited similar failure modes across multiple tasks and prompts – indicating limited capacity for planning. In contrast, Gemini offers more advanced high-level reasoning, which is central to our framework. Additionally, recent versions of Gemini have also demonstrated strong performance in spatial grounding and pointing, making it particularly well-suited for scaffold generation in our setting. One could continue running experiments to determine the best VLM for each component of the pipeline, but we view doing so as further optimization which would only improve performance.
>
> # Impact of VLM Prediction Errors
>
> We agree that the effectiveness of our method is influenced by the accuracy of outputs from the VLM. Errors in keypoint detection or trajectory prediction can indeed degrade downstream RL  performance.
>
> To quantify this, we have already included results using oracle keypoint and trajectory predictions – providing an upper bound on performance in the presence of perfect VLM outputs. These are shown in Figure 7 (right) as “Keypoint Oracle” and “Trajectory Oracle.” As discussed in the paper, oracle predictions yield varying degrees of performance improvement across different tasks, and in some cases, enable near-perfect success rates.
>
> As requested, we have added results in the reverse direction -- demonstrating how performance degrades with artificially injected gaussian noise. Specifically, we add Gaussian noise N(0, sigma^2) to the waypoints proposed by the VLM, thereby simulating planning errors. We add noise both during training and inference. We report success rates as the mean +- standard error computed over three random seeds for each task.
> | method              | Apple       | Drawer      | Hammer      | Scissors    |
> |---------------------|-------------|-------------|-------------|-------------|
> | No noise (sigma: 0) | 0.84 ± 0.01 | 0.87 ± 0.01 | 0.67 ± 0.01 | 0.67 ± 0.01 |
> | Noisy (sigma: 0.01) | 0.58 ± 0.29 | 0.88 ± 0.00 | 0.72 ± 0.01 | 0.57 ± 0.02 |
> | Noisy (sigma: 0.02) | 0.84 ± 0.01 | 0.87 ± 0.00 | 0.72 ± 0.02 | 0.13 ± 0.01 |
> | Noisy (sigma: 0.05) | 0.36 ± 0.16 | 0.86 ± 0.01 | 0.24 ± 0.03 | 0.34 ± 0.08 |
>
> # Limitations of Representation
>
> Note that motion scaffolds are designed to 1) provide dense rewards for the agent and 2) guide exploration with residual actions. This does not limit the information that the RL policy can learn from – it could use force information or contacts if desired to learn to do the task effectively though we did not as such information is more difficult in the real world. However, there is nothing about our framework that prevents the policy from doing so. While our reward function is based on spatial information, the low-level RL agent can explore different amounts of force, torque, and contact behaviors. For example, we found that the RL agent automatically used stable grasps with high contact forces for pick and place tasks with large objects, while choosing small forces for finer tasks such as closing the pair of scissors.
>
> On the other hand, adding force and torque constraints to the reward function is possible, but doing so requires an extensive amount of manual reward design which may not be broadly applicable. The purpose of our work, however, was to show how much scaffolds reduce the burden of reward design across a broad set of tasks. To this end, while adding more specific information could increase performance in specific cases, it comes at the cost of generality. If a task requires additional terms, they could be added on top of the terms for the motion scaffold and used in combination. We leave this exploration to future work.
>
> Furthermore, the spatial information encoded by our reward design can go really far. To demonstrate this, we modified the original bottle task: the bottle may now start either lying down or standing upright on one side of a sink and must be placed upright on the opposite side. This setup requires the policy to infer the initial pose and adjust its behavior accordingly to rotate the bottle upright – all from using multiple keypoints to infer rotation. Unfortunately, due to the limitations of rebuttal this year we are unable to attach images or videos. We report results across three seeds below:
>
> | Method               | Success Rate (mean ± SE) |
> |----------------------|--------------------------|
> | Pre-recorded baseline | 0.09 ± 0.02              |
> | Ours (zero-shot)      | 0.64 ± 0.10              |
>
> While the discussion above focused primarily on reward design, we would like to note that there is a plethora of existing literature in constrained RL [Responsive Safety in RL by PID Lagrangian Methods, Stooke et al 2020, Constrained RL with Smoothed Log Barrier Function Zhang et al 24], which is orthogonal to our effort which could be used for actually enforcing constraints instead of “encouraging compliance” with reward penalties.

---

> > ### Comment · Reviewer_5ndj · 2025-08-05
> > **Response**
> >
> > I would like to thank the author for the response, however based on the rebuttal points, i would like to retain my rating.

---

### Official Review · Reviewer_a8sd · 2025-07-03

**Clarity:** 3
**Significance:** 2
**Originality:** 2
**Rating:** 4
**Confidence:** 4

**Summary:**

This paper presents a two-stage method for dexterous manipulation. An off-the-shelf VLM serves as a high-level planner that generates keypoints or waypoints and synthesizes 3D trajectories, while a reinforcement-learning policy in simulation tracks the coarse trajectories. The experiments cover an eight-task suite in simulation, comparisons with several baselines, failure analysis, and ablation studies on the number of few-shot examples and waypoints. The Place Bottle and Slide Box tasks are also deployed on real robots.

**Questions:**

See weaknesses.

**Ethical Concerns:**

["NO or VERY MINOR ethics concerns only"]

**Final Justification:**

Considering the new baselines and tasks, I will increase my score to borderline accept

**Limitations:**

Yes.

**Quality:**

3

**Strengths And Weaknesses:**

**Strengths:**

1. Combining VLM and RL in simulation is reasonable. It shows that VLM-generated trajectories can solve articulation and precise manipulation tasks (although these are in simulation only, not real world).
2. The simulation task suite contributes to the community.
3. The failure analysis provides useful insights.
4. The presentation quality is high; figures and writing are well polished.

**Weaknesses:**

1. The method is not compared with strong baselines. All baselines in Fig. 4 are self-crafted (not compared with SOTA methods mentioned in Section 2 "Related Work" such as RoboDexVLM and Iterative Keypoint Rewards). It is unclear how the proposed method differs from related works, and more importantly, whether it performs better.
2. The RL is trained in simulation, which assumes access to a real-to-sim digital twin. This limits generalizability to new tasks and environments. Reproducing the experiment requires building a real-to-sim setup, which is not well discussed in the paper.
3. Why are only simpler tasks deployed in the real world? What about articulation and precise manipulation, which are only shown in simulation?
4. How is kinematic reachability handled when deploying in the real world? In simulation, a free-floating hand is used. Would there be many invalid trajectories generated by the VLM or RL due to this simplification?

---

> ### Author Rebuttal · Authors · 2025-07-31
>
> We thank the reviewers for their constructive feedback. Below, we address their stated concerns.
> # Baseline Comparisons
> To strengthen the empirical evaluation, we have incorporated four additional baselines. We report success rates as the mean +- standard error computed over three random seeds for each task.
> ## Iterative Keypoint Rewards (IKER)
> We implement an IKER-style baseline in which a vision-language model (VLM) generates code for reward function parameters, using the exact prompts and methodology from the original paper. However, as IKER assumes a fixed set of keypoints, we use the same VLM-identified keypoints as in our method for parity.
> | method           | Apple       | Bottle      | Drawer      | Fridge      | Hammer      | Pliers      | Scissors    | Sponge      |
> |------------------|-------------|-------------|-------------|-------------|-------------|-------------|-------------|-------------|
> | IKER (zero-shot) | 0.45 ± 0.14 | 0.63 ± 0.08 | 0.23 ± 0.01 | 0.19 ± 0.04 | 0.53 ± 0.03 | 0.19 ± 0.11 | 0.27 ± 0.03 | 0.06 ± 0.02 |
> | Ours (zero-shot) | 0.84 ± 0.01 | 0.88 ± 0.00 | 0.87 ± 0.01 | 0.12 ± 0.00 | 0.67 ± 0.01 | 0.80 ± 0.03 | 0.67 ± 0.01 | 0.87 ± 0.02 |
>
> ## RL From Scratch
> We introduce an additional baseline in which a policy is trained from scratch using RL. The reward function mirrors that of our method – combining contact and keypoint-distance-to-target – but relies on oracle keypoints rather than those detected by a VLM. Importantly, this baseline does not use any trajectory guidance or demonstrations. To ensure a fair comparison, we evaluate it alongside our method using the same oracle keypoints (i.e., Ours + keypoint oracle), but retaining VLM-generated trajectories. Policies are trained using PPO.
>
> | method                      | Apple       | Bottle      | Drawer      | Fridge      | Hammer      | Pliers      | Scissors    | Sponge      |
> |-----------------------------|-------------|-------------|-------------|-------------|-------------|-------------|-------------|-------------|
> | Simple RL + keypoint oracle | 0.00 ± 0.00 | 0.99 ± 0.00 | 0.00 ± 0.00 | 0.00 ± 0.00 | 0.00 ± 0.00 | 0.65 ± 0.33 | 1.00 ± 0.00 | 0.00 ± 0.00 |
> | Ours + keypoint oracle      | 0.90 ± 0.02 | 0.85 ± 0.02 | 0.82 ± 0.00 | 0.26 ± 0.01 | 0.81 ± 0.01 | 0.85 ± 0.01 | 0.91 ± 0.02 | 0.96 ± 0.01 |
>
> ## RL with Complex Reward Design
> We also introduce a baseline that employs a more detailed, task-specific, hand-crafted reward function. This reward includes contact, keypoint-distance-to-target, hand-distance-to-object, and an additional success signal, assuming access to a success detector. This baseline is intended to reflect the upper bound of performance achievable through extensive manual reward engineering. As with the previous baseline, it does not utilize any trajectory supervision or demonstrations. To ensure a fair comparison, we evaluate it alongside our method using the same oracle keypoints (Ours + keypoint oracle). Policies are trained using PPO.
> | method                       | Apple       | Bottle      | Drawer      | Fridge      | Hammer      | Pliers      | Scissors    | Sponge      |
> |------------------------------|-------------|-------------|-------------|-------------|-------------|-------------|-------------|-------------|
> | Complex RL + keypoint oracle | 0.98 ± 0.01 | 0.99 ± 0.00 | 1.00 ± 0.00 | 0.00 ± 0.00 | 0.00 ± 0.00 | 0.99 ± 0.00 | 1.00 ± 0.00 | 0.00 ± 0.00 |
> | Ours + keypoint oracle       | 0.90 ± 0.02 | 0.85 ± 0.02 | 0.82 ± 0.00 | 0.26 ± 0.01 | 0.81 ± 0.01 | 0.85 ± 0.01 | 0.91 ± 0.02 | 0.96 ± 0.01 |
>
> ## Imitation Learning (IL)
> To evaluate imitation learning, we train Diffusion Policy using successful rollouts of our oracle RL policy as demonstrations. We conduct experiments using 10, 20, and 50 demonstrations. This represents the best-case performance for imitation learning: the demonstrations are perfect, resulting from scripted trajectories and an RL policy, unlike standard human demonstrations.
>
> | method                      | Apple       | Bottle      | Drawer      | Fridge      | Hammer      | Pliers      | Scissors    | Sponge      |
> |-----------------------------|-------------|-------------|-------------|-------------|-------------|-------------|-------------|-------------|
> | Diffusion Policy (10 demos) | 0.26 ± 0.04 | 0.07 ± 0.01 | 0.47 ± 0.03 | 0.36 ± 0.29 | 0.45 ± 0.11 | 0.45 ± 0.10 | 0.52 ± 0.08 | 0.94 ± 0.03 |
> | Diffusion Policy (20 demos) | 0.52 ± 0.02 | 0.17 ± 0.01 | 0.74 ± 0.10 | 0.60 ± 0.20 | 0.45 ± 0.07 | 0.60 ± 0.15 | 0.70 ± 0.10 | 0.99 ± 0.00 |
> | Diffusion Policy (50 demos) | 0.84 ± 0.02 | 0.80 ± 0.05 | 0.85 ± 0.08 | 0.96 ± 0.02 | 0.65 ± 0.09 | 0.79 ± 0.08 | 0.89 ± 0.02 | 0.99 ± 0.00 |
> | Ours (few-shot)             | 0.87 ± 0.02 | 0.86 ± 0.04 | 0.92 ± 0.01 | 0.64 ± 0.04 | 0.75 ± 0.00 | 0.86 ± 0.02 | 0.76 ± 0.03 | 0.84 ± 0.05 |
> | Ours (zero-shot)            | 0.84 ± 0.01 | 0.88 ± 0.00 | 0.87 ± 0.01 | 0.12 ± 0.00 | 0.67 ± 0.01 | 0.80 ± 0.03 | 0.67 ± 0.01 | 0.87 ± 0.02 |
>
>
> We do not compare against RoboDexVLM because their method relies on a library of manually crafted primitive skills, while our method tries to learn those skills implicitly with RL.  Those primitives are difficult to scale to unstructured motions such as hammering, or precise skills such as closing a pair of scissors. Because of this, RoboDexVLM sticks to simpler grasping tasks. Additionally, the authors of RoboDexVLM have not released their code, making replication difficult.
> # Real-to-Sim Transfer
> We acknowledge that our approach assumes a real-to-sim transfer pipeline. However, we would like to note that this is a common assumption made by works in dexterous manipulation [ObjDex Chen et al. 24, RoboDexVLM Liu et al. 25, Solving a Rubrik’s Cube, OpenAI et al. 19]. To our knowledge there are only a handful of works that train dexterous RL policies entirely in the real world, which comes with its own challenges (resets etc.) [PDDM, Nagabandi et al. 19]. While we do assume access to a simulated model of the robot, such an assumption is commonplace in several renowned works in dexterous manipulation. We will add more details about our specific sim-to-real pipeline in the final version of the manuscript.
> # Real-World Task Complexity
> We appreciate the suggestion to explore more challenging tasks in the real world. In response, we have added a **hammering** task, where the robot grasps a hammer by the handle and performs three strikes on a table. This task requires substantially more precision than previous ones (bottle-picking or pushing) and our method achieves a 60% success rate out of 20 trials. Due to the limitations of the rebuttal this year, we are unfortunately unable to upload links or images. To our knowledge, this is the first demonstration of such a task using a multi-fingered robot hand trained via RL.
>
> For more precise tasks, we are currently limited by hardware constraints, including depth sensor resolution [Learning to estimate 3d states of deformable objects from single-frame occluded point clouds, Lv et al 23] and the mechanical tolerances of the Allegro Hand [LEAP Hand, Shaw et al RSS 23] which restrict performance on high-precision manipulation.
> # Kinematic Reachability
> For all real-world experiments, we use a 7-DoF robotic arm in both simulation and the physical setup. This ensures that the policy is trained with kinematic constraints similar to those of the real world. The floating-hand configuration is used only for the simulation-only experiments. In both setups, the policy outputs 3D wrist poses, which are converted to joint angles via the pseudo-inverse method. We ensure that the initial hand pose is kinematically feasible and that all objects are placed within the robot’s reachable workspace.
>
> While VLMs are initially unaware of robot kinematics, our few-shot refinement phase conditions the model on successful prior samples, improving its ability to generate feasible trajectories. Additionally, we introduce substantial noise during plan sampling, encouraging diversity of action proposals such that a significant number of the sampled plans during RL training are kinematically viable.

---

> > ### Author Response · Authors · 2025-08-05
> >
> > Dear reviewer,
> >
> > We would like to follow up to see if the response addresses your concerns or if you have any further questions. We would really appreciate the opportunity to discuss this further if our response has not already addressed your concerns. Thank you again!

---

> > ### Comment · Reviewer_a8sd · 2025-08-06
> >
> > Thanks for the new baselines; they are more convincing compared to the original submission. Also, thank you for the new hammering task and the details on Kinematic Reachability.
> >
> > However, regarding the real-to-sim transfer: while it is common to train RL in simulation, it is more common to perform zero-shot VLM planning directly in the real world, as done in ReKep, Prompting with the Future, and VoxPoser.
> >
> > That said, considering the new baselines and tasks, I will increase my score.

---

> > > ### Author Response · Authors · 2025-08-06
> > > **Clarification on Sim2Real Pipeline**
> > >
> > > Hi, thanks for the response! We would just like to clarify that the VLM planning is indeed done in the real world when rolling out the policy in real. While the low-level RL policy is trained in sim on key-points and trajectories generated in simulation, when transferring the policy to the real world we re-run the VLM prompt in the real world to identify key-points and use images of the real environment to generate the motion scaffold.

---

### Note · Authors · 2025-08-16

We thank the reviewers for their thoughtful and constructive feedback. Below we summarize the main concerns raised and how we addressed them.

## 1. Limited Comparisons

**Concern**: Our approach was not compared against sufficiently strong baselines.

**Response**: We added four baselines:

- Iterative Keypoint Rewards (IKER): Our method consistently outperforms IKER across nearly all tasks.
- RL from scratch with oracle keypoints: Solves only a subset of tasks and relies on privileged state information.
- RL with complex hand-crafted rewards: Strong in many tasks but heavily dependent on task-specific engineering and success detectors.
- Diffusion Policy with 10/20/50 perfect demonstrations: Our method performs similarly to 50-demo Diffusion Policy, despite requiring no demonstrations.

Together, these results show scaffold-based supervision is competitive with, and often superior to, state-of-the-art methods while being substantially more practical.

## 2. Real-World Complexity

**Concern**: The real-world experiments were too simple.

**Response**: We added a more demanding hammering task, requiring the robot to grasp a hammer by the handle and perform repeated strikes. This requires higher precision and more complex planning. Our method achieved 60% success over 20 trials, with failures mainly due to hardware limitations.

## 3. Generality of Instructions

**Concern**: Prompts may encode task- or environment-specific knowledge.

**Response**: We now use a single concise instruction per task. This instruction is embedded into generic prompts. Performance remained stable, showing that the method generalizes effectively without relying on specialized or hidden priors.

## 4. Lack of Explicit Orientation Information

**Concern**: The method may not succeed on orientation-dependent tasks.

**Response**: We modified the bottle pick-and-place task so the bottle could start upright or flat but must end upright. Using multiple keypoints, the method inferred orientation successfully, showing that scaffolds extend naturally to orientation-dependent manipulation.

## Conclusion

Our method enables dexterous, contact-rich manipulation without demonstrations or reward engineering, and scales to realistic tasks. It performs competitively against baselines that often require substantially more resources, such as demonstrations or privileged information. Scaffold-based supervision thus offers a practical and scalable foundation for advancing dexterous robot learning.

---

### Decision · Program_Chairs · 2025-09-17

**Decision:**

Accept (poster)

**Comment:**

This paper proposes a two-stage framework for dexterous manipulation: a high-level VLM module that produces semantic keypoints and trajectories, and a low-level reinforcement learning policy that tracks and refines them. The system is evaluated on an eight-task suite in simulation and several real-world tasks, with ablations, baseline comparisons, and a new hammering task added in the rebuttal.

**Strengths**

- The idea of using VLM-generated motion scaffolds as lightweight supervision for RL is novel and impactful. It sidesteps the reliance on expensive demonstrations or hand-engineered rewards, thereby reducing human effort.
- The modular design is attractive: the VLM can be upgraded, and the residual RL policy naturally complements the coarse guidance.
- The work includes a thorough set of experiments, including new baselines (IKER, RL from scratch, imitation learning with diffusion policies, and complex reward engineering) that position the method more convincingly relative to prior art.
- The presentation quality is consistently high, with polished writing, clear figures, and informative failure analyses.
- Real-world deployment, especially the added hammering task, strengthens the contribution and shows feasibility beyond simulation.

**Weaknesses**

- Concerns remain about the realism and generality of the approach. Several reviewers noted that simulated motions (e.g., hammering, pliers) appear physically unrealistic, raising questions about whether coarse scaffolds are sufficient for complex dexterous control.
- The reliance on a real-to-sim pipeline limits applicability, and while the authors clarify that VLM planning is re-run in the real setting, the necessity of digital twins remains a barrier to broader deployment.
- Orientation and force/torque information are not explicitly represented in the scaffold, which constrains performance in precision tasks. The authors partially address this with multi-keypoint orientation inference, but success rates degrade in harder reorientation tasks.
- Prompting remains somewhat task- and environment-specific, and while the rebuttal shows improvements with standardized prompts, generalization to unseen tasks and scenes is still not fully resolved.
- From a technical standpoint, individual components (VLM planning, residual RL, scaffold-based rewards) are adaptations of existing ideas rather than fundamentally new algorithms. The contribution is primarily an engineering integration, though one that is creative and useful.

**Overall Assessment**

Reviewers converged on a borderline but leaning accept consensus. Some (e.g., R5ndj, R5Gyk) valued the system-level novelty and engineering contribution, while others (e.g., Rb94o) remained unconvinced about physical realism and generalization. The rebuttal added meaningful new evidence: stronger baselines, more challenging real-world demonstrations, and clarifications on kinematic reachability and scaffold orientation. These updates improve confidence in the technical soundness of the approach, even if limitations remain.

**Recommendation**

On balance, the paper presents an interesting and well-executed idea with clear relevance to the NeurIPS robotics community. Despite limited technical novelty and lingering concerns about sim-to-real fidelity, the integration of VLM planning with RL for dexterous control is interesting and worth to be presented to the community. As a result, the ACs recommend acceptance.